# Hippo signaling differentially regulates distal progenitor subpopulations and their transitional states to construct the mammalian lungs

Kuan Zhang[1,2,4], Madhuri Basak [1,4], Youssef Zaher[1,4], Erica Yao [1], Shao-An Wang [1,3], Thin Aung[1] & Pao-Tien Chuang [1] ✉

Lung size control and cell type specification are key unresolved issues. In this study, we have engineered mosaic patterns of Hippo signaling to reveal the developmental potential of SOX9[+] progenitors at the distal lung buds. Our results show that the distal SOX9[+] subdomain is sufficient to direct lung outgrowth through bifurcation, providing a mechanism for lung size control. Through single-cell analyses, we identify transitional cell states and candidates for promoting cell fates. Moreover, genetic analysis reveals that Hippo signaling induces distinct cell fates at different SOX9[+] subdomains to produce the conducting airways and the alveolar epithelium. These results provide the first extensive map of the developmental paths of lung cells. Some of the developmental paths of transitional cell states in mice correspond to those in human lungs. Together, these studies provide mechanistic insight into how Hippo signaling controls the sequential expansion and differentiation of SOX9[+] progenitors to construct the mammalian lungs.

Execution of a stereotypical branching program during lung development generates an elaborate respiratory tree. Following branching, the distal ends of lung branches enlarge to form saccules, the precursor of alveoli. During these developmental processes, distinct lung cell types that carry out requisite physiological functions are produced at designated locations. This design enables mucociliary clearance in the airways and gas exchange in the alveoli. The time sequence of morphological changes of the developing lung has been characterized[1–4]. However, a mechanistic understanding of lung development and cell differentiation remains incomplete[4–6].

Lineage tracing and transcriptome analysis have led to a model in which a SOX9[+] progenitor pool located at the distal tip of lung branches gives rise to all lineages in the lung except for the trachea and main-stem bronchi[2,7]. In this model, distal SOX9[+] progenitors first produce SOX2[+] cells, which subsequently give rise to lung cells in the

conducting airways. Later in lung development, distal SOX9[+] progenitors produce alveolar type 1 (AT1) and alveolar type 2 (AT2) cells, essential components of the alveoli[7]. A main deficiency of this model is the lack of functional data on SOX9[+] progenitors at different subdomains. It also does not provide a mechanistic description of how to maintain a SOX9[+] progenitor pool of proper size and how to induce differentiation of SOX9[+] cells into either the conducting or alveolar cell fates along the proximo-distal axis of lung branches. In this study, we employed the Hippo signaling pathway as a tool to gain molecular insight into these key issues.

The Hippo pathway was discovered for its role in controlling organ size[8]. Hippo signaling regulates fundamental cellular processes critical to tissue growth and regeneration. The basic framework of Hippo signal transduction has been elucidated[9–15]. In response to external cues or high cell density, a kinase cascade leads to the

[1]Cardiovascular Research Institute, University of California, San Francisco, CA, USA. [2]School of Biomedical Science, Hunan University, Changsha, China. [3]School of Respiratory Therapy, College of Medicine, Taipei Medical University, Taipei, Taiwan. [4]These authors contributed equally: Kuan Zhang, Madhuri Basak, Youssef Zaher. ✉e-mail: pao-tien.chuang@ucsf.edu

phosphorylation of the transcription coactivators, YAP (yes-associated protein) and TAZ (transcriptional coactivator with PDZ-binding motif), which are sequestered in the cytoplasm and degraded. In this process, STK3/4 (*i.e.*, MST1/2 [mammalian sterile 20-related 1 and 2 kinases]) phosphorylate LATS1/2 (large tumor suppressor kinase 1 and 2), which in turn phosphorylate YAP (denoted as phospho-YAP or pYAP) and TAZ. When cell density is low, STK3/4 and LATS1/2 kinases are inactive. Unphosphorylated YAP and TAZ enter the nucleus and bind to the TEAD transcription factor to activate target gene expression and influence many cellular functions. Among them are cell proliferation, cell migration, cell differentiation, and cell apoptosis.

Analysis of mouse lungs deficient in *Stk3/4* (*Mst1/2*), *Lats1/2*, and *Yap/Taz* has been reported[16–20]. These studies established an essential role of Hippo signaling in lung development. Nevertheless, the molecular mechanisms by which Hippo signaling controls the distinct steps of expansion and differentiation of SOX9+ progenitors remain unclear. A main hurdle is the inability to manipulate Hippo signaling in a subset of SOX9+ progenitors at different locations for phenotypic and molecular analysis.

In this study, we have utilized mosaic or regional inactivation of *Yap/Taz* and *Lats1/2* by Cre/CreER lines in different subdomains of SOX9+ progenitors to reveal the role of Hippo signaling. The phenotypic analysis is coupled with transcriptome and chromatin accessibility assays, which identify distinct transitional states and their regulators. Together, our findings provide mechanistic insight into how the SOX9+ progenitor pool enables the construction of the respiratory tree and the production of its relevant cell types.

## Results

### Hippo signaling regulates the size of the distal SOX9+ domain of lung branches in mice

Branching defects in *Yap*- and *Lats1/2*-deficient lungs have been reported[16,18–20]. Although these studies suggest that the distal SOX9+ domain is regulated by Hippo signaling, they failed to reveal how YAP/TAZ levels affect the SOX9+ progenitor pool at the molecular level.

*Yap^f/f*; *Shh^{Cre/+}* and *Yap^f/f*; *Sox9^{Cre/+}* lungs, in which *Shh^{Cre}*[21] or *Sox9^{Cre}*[22] converted a floxed allele of *Yap* (*Yap^f*)[23] into a null allele, are characterized by distal lung cysts. While *Shh^{Cre}* is broadly expressed in lung epithelial cells, *Sox9^{Cre}* is mainly restricted to distal SOX9+ cells. We found that lung cysts were lined by SOX9+ cells and lacked SOX2+ cells. SOX2+ cells are derived from SOX9+ cells, located proximally to SOX9+ cells, and will produce the conducting airways. This suggests that SOX2+ cells failed to be produced in *Yap*-deficient lung epithelium (Supplementary Fig. 1). Moreover, the number of SOX9+ cells appeared to be reduced in the mutant lung (Supplementary Fig. 1a–d).

Unexpectedly, activation of YAP/TAZ through the removal of *Lats1/2* also resulted in a smaller SOX9+ domain (Supplementary Fig. 1e–h). The LATS1/2 kinases phosphorylate YAP, leading to the sequestration and degradation of pYAP in the cytoplasm. Thus, loss of *Lats1/2* would block YAP degradation and activate YAP targets. *Lats1^f/f*; *Lats2^f/f*; *Shh^{Cre/+}* mice have been reported[19], in which the epithelial *Shh^{Cre}* converted floxed alleles of *Lats1* (*Lats1^f*) and *Lats2* (*Lats2^f*)[24,25] into null alleles.

Together, these studies support a model in which Hippo signaling regulates the balance between the proliferation and differentiation of SOX9+ progenitors. We propose that YAP/TAZ levels need to be tightly regulated to prevent depletion of the SOX9+ progenitor pool due to either an undersized pool (low YAP/TAZ) or precocious differentiation (high YAP/TAZ), respectively. We will further test this model in this study.

### YAP/TAZ levels regulate lung cell differentiation in mice

To test whether a SOX9+ progenitor pool provides the source for lung growth and cell differentiation, we activated YAP/TAZ in the lung epithelium through *Lats1/2* removal. The removal of *Lats1/2* by *Shh^{Cre}* failed to reveal the direct effects of YAP/TAZ on SOX9+ progenitors due

to the early and broad expression of *Shh^{Cre}*. Moreover, we found that mouse embryos in which *Lats1/2* were inactivated by *Sox9^{Cre}*, another mouse epithelial Cre line, died prior to the initiation of lung development and were uninformative. To overcome a lack of precise spatial Cre activity in the lung epithelium, we took advantage of *SPC^{Cre}* (i.e., *Sftpc^{Cre}*), in which Cre expression under the *surfactant protein C* (SPC or SFTPC) promoter exhibited non-uniform conversion of floxed alleles to null alleles in the lung epithelium, as we previously reported[18].

*Lats1^f/f*; *Lats2^f/f*; *Sftpc^{Cre/+}* mice (denoted as *Lats1/2*-mosaic mice in this study) were born alive but died soon after birth. The lungs of most *Lats1/2*-mosaic mice were devoid of air (Fig. 1a, b). The lung size of *Lats1/2* mosaic mice was similar to or slightly less than that of their littermate controls (Fig. 1c).

Lung branching was disrupted (Fig. 1d, e) and marker expression for cells in the conducting airways was diminished in *Lats1/2*-mosaic lungs (Fig. 1f, g). For instance, the number of secretory club (Clara) cells (SCGB1A1+ [CC10+]) and ciliated cells (Ac-tub+) was greatly decreased in *Lats1/2*-mosaic lungs compared to controls (Fig. 1f, g). This indicates that not only did proper branching fail to occur in *Lats1/2*-deficient descendants, but elevated YAP/TAZ levels also disrupted the production of cell types in the conducting airways. This is consistent with a model in which the immediate progeny (the transition progenitors) of the SOX9+ tip progenitors[7] are the source for constructing the conducting airways.

Saccules, which are enlargements of the terminal ends of lung branches, failed to form in the mutant lungs (Fig. 1h, i). *Lats1/2*-deficient cells (SOX9−SOX2−) (details below) that occupied the disorganized airways in *Lats1^f/f*; *Lats2^f/f*; *Sftpc^{Cre/+}* (*Lats1/2*-mosaic) lungs expressed AT1 markers (*e.g.*, HOPX) (Fig. 1j–m). Fibroblasts/myofibroblasts (PDGFRA+) and vascular smooth muscle cells (PDGFRB+) were interspersed with AT1 cells (HOPX+) (Fig. 1n, o), mimicking the interactions between these cell types in the developing saccules/alveoli. By contrast, few, if any, cells expressed AT2 markers (*e.g.*, SFTPC) in *Lats1/2*-mosaic lungs (Fig. 1j–m). These results suggest that high YAP/TAZ activity converts SOX9+ or SOX2+ cells into the SOX9−SOX2− state, which then turns on the AT1 fate.

### Yap/Taz mediate Lats1/2 function in regulating murine lung development

The Hippo pathway is known for its context-dependent signaling. The signaling network that involves Hippo signaling may differ in various tissues and organs. Moreover, kinases other than STK3/4 and LATS1/2 can participate in the network[26], leading to YAP/TAZ phosphorylation. It was thus uncertain if the lung phenotypes in *Lats1/2*-deficient mice were due to increased YAP/TAZ levels. To resolve this issue, we generated *Lats1^f/f*; *Lats2^f/f*; *Yap^f/+*; *Taz^f/+*; *Shh^{Cre/+}* mice, in which one copy of *Yap* and *Taz* was removed in *Lats1/2*-deficient lungs by *Shh^{Cre}*. Namely, a floxed allele of *Yap* (*Yap^f*) or *Taz* (*Taz^f*)[27] was converted into a null allele. Analysis of lungs from *Lats1^f/f*; *Lats2^f/f*; *Yap^f/+*; *Taz^f/+*; *Shh^{Cre/+}* mice revealed normal branching and proper production of lung cell types at the correct positions, in a pattern similar to that of control lungs (Supplementary Fig. 2). These results suggest that *Lats1/2* function through *Yap/Taz* in controlling the SOX9+ progenitors. Surprisingly, *Lats1^f/f*; *Lats2^f/f*; *Yap^f/+*; *Taz^f/+*; *Shh^{Cre/+}* mice did not survive after birth, suggesting that the rescue was partial. It is unclear if some aspects of pulmonary and/or extrapulmonary function were not fully restored.

We also produced control and *Lats1^f/f*; *Lats2^f/f*; *Yap^f/+*; *Taz^f/+*; *Sftpc^{Cre/+}* mice. Analysis of lungs from *Lats1^f/f*; *Lats2^f/f*; *Yap^f/+*; *Taz^f/+*; *Sftpc^{Cre/+}* mice disclosed partial rescue of the branching defects (Fig. 2a–c) and cell type specification (Fig. 2d–j). This reinforced the notion that *Sftpc^{Cre}* has a lower efficiency in eliminating *Yap* and *Taz* than *Shh^{Cre}*. *Lats1^f/f*; *Lats2^f/f*; *Yap^f/+*; *Taz^f/+*; *Sftpc^{Cre/+}* mice also failed to survive after birth.

Together, these results support the idea that *Lats1/2* function through *Yap/Taz* to control the number and differentiation of SOX9+

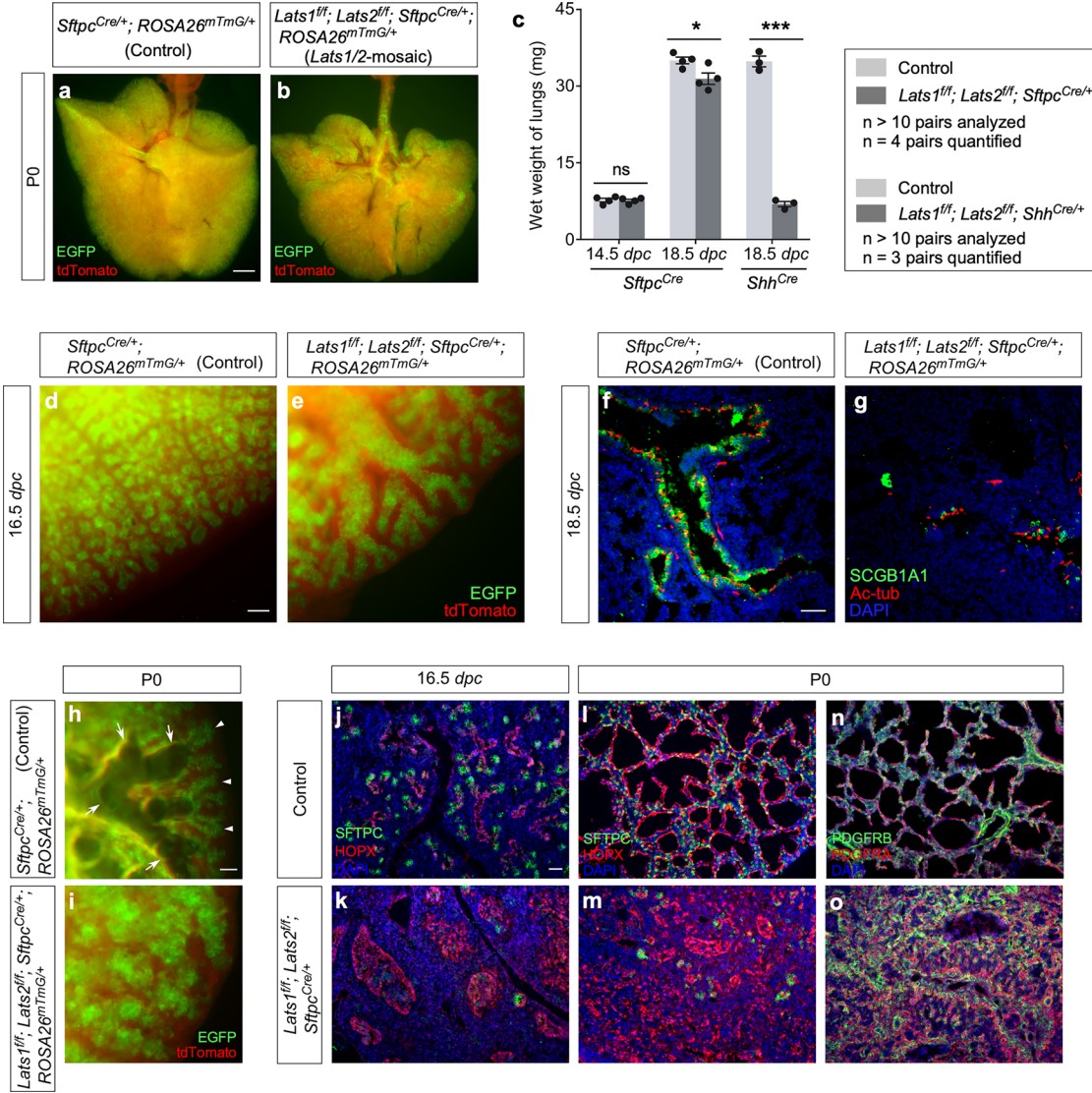

**Fig. 1 | The distal SOX9⁺ progenitors control lung size. a, b** Whole-mount imaging of lungs (ventral view) from *Sftpc^{Cre/+}; ROSA26^{mTmG/+}* (Control) and *Lats1^{f/f}; Lats2^{f/f}; Sftpc^{Cre/+}; ROSA26^{mTmG/+}* (*Lats1/2*-mosaic) mice at postnatal (P) day 0. The EGFP signal from the *ROSA26^{mTmG}* allele was induced by *Sftpc^{Cre}*. It has been noted that EGFP expression from *ROSA26^{mTmG}* reporter mice can be induced by low levels of Cre, at which conversion of floxed alleles into null alleles may not occur. While all epithelial cells in *Lats1^{f/f}; Lats2^{f/f}; Sftpc^{Cre/+}; ROSA26^{mTmG/+}* lungs were labeled by EGFP (green), removal of epithelial *Lats1/2* by *Sftpc^{Cre}* was mosaic. These findings support the notion that the mosaic nature of *Sftpc^{Cre}* originates from its selective (regional) effects on converting floxed alleles of *Lats1/2* into null alleles and not from selective (regional) expression of *Sftpc^{Cre}*. **c** Measurement of the wet weight of lungs from control and *Lats1^{f/f}; Lats2^{f/f}; Sftpc^{Cre/+}* (*n* = 4 pairs), and control and *Lats1^{f/f}; Lats2^{f/f}; Shh^{Cre/+}* mice (*n* = 3 pairs) at 14.5 and 18.5 *days post-coitus* (*dpc*). The wet weight of *Lats1^{f/f}; Lats2^{f/f}; Sftpc^{Cre/+}* lungs was only slightly less than that of control lungs at 18.5 *dpc*. By contrast, the wet weight of *Lats1^{f/f}; Lats2^{f/f}; Shh^{Cre/+}* lungs was a small fraction of that of control lungs. Of note, loss of one, two, or three alleles of *Lats1/2* by *Sftpc^{Cre}* or *Shh^{Cre}* did not result in any apparent lung defects, consistent with a functional redundancy between *Lats1* and *Lats2*. **d, e** Whole-mount imaging of dissected lungs from *Sftpc^{Cre/+}; ROSA26^{mTmG/+}* (Control), *Lats1^{f/f}; Lats2^{f/f}; Sftpc^{Cre/+}; ROSA26^{mTmG/+}* (*Lats1/2*-mosaic) mice at 16.5 *dpc*. The images showed a surface view of the distal part of the lung, where the edge of the lung was visible. EGFP was activated from the *ROSA26^{mTmG}* allele in almost all lung epithelial cells except those in the trachea; tdTomato marked all non-epithelial cells where *Sftpc^{Cre}* was not expressed. **f, g** Immunostaining of lung sections collected from control and *Lats1/2*-mosaic mice at 18.5 *dpc*. SCGB1A1 (CC10) marked club (Clara) cells; acetylated

tubulin (Ac-tub) marked ciliated cells in the airways. **h, i** Whole-mount imaging of lungs from control and *Lats1/2*-mosaic lungs at P0. The branching pattern was disrupted, and saccules failed to form in *Lats1/2*-mosaic lungs. Arrows point to the open saccules filled with air, while arrowheads point to the closed saccules in the control lungs. **j–o** Immunostaining of lung sections collected from control and *Lats1/2*-mosaic mice at 16.5 *dpc* and P0. SFTPC (green) labeled alveolar epithelial type 2 (AT2) cells, and HOPX (red) detected alveolar epithelial type 1 (AT1) cells. PDGFRA (red) marked fibroblasts/myofibroblasts, and PDGFRB (green) labeled pericytes and vascular smooth muscle cells. The *Lats1/2*-mosaic lungs were distinguished by a stratified airway epithelium, predominantly composed of AT1 cells. At 16.5 *dpc*, SFTPC⁺ cells were located at the very tip of the lung epithelium while HOPX⁺ cells occupied the immediately adjacent position proximal to SFTPC⁺ cells, as shown in (**j**). *Lats1/2*-mosaic lungs showed a disorganized lung epithelium that expressed only HOPX (**k**). At P0, interestingly, HOPX⁺ cells in *Lats1/2*-mosaic lungs appeared to segregate from each other, interspersed with HOPX⁻ cells (**m**) expressing PDGFRA⁺ (**o**). We propose that persistent proliferation at the distal region where *Lats1/2* were removed underlies the disorganized epithelium in *Lats1/2*-mosaic lungs. *Lats1/2*-deficient cells lost their SOX9⁺ progenitor property and failed to execute domain branching while proliferating. Instead of forming branches along the stalk, *Lats1/2*-deficient cells aggregate, forming a disorganized lung epithelium. We also speculate that folding/flattening of HOPX⁺ cells contributes to the generation of a disorganized epithelium in *Lats1/2*-mosaic lungs and is, in part, induced by infiltrating alveolar PDGFRA⁺ fibroblasts. All values are mean ± SEM. (*) *p* < 0.05; (***) *p* < 0.001; ns, not significant (two-tailed Student's *t*-test). Scale bars, 1 mm (**a, b**), 100 µm (**d, e**), 50 µm (**f, g**), 25 µm (**h, i**), 100 µm (**j–o**).

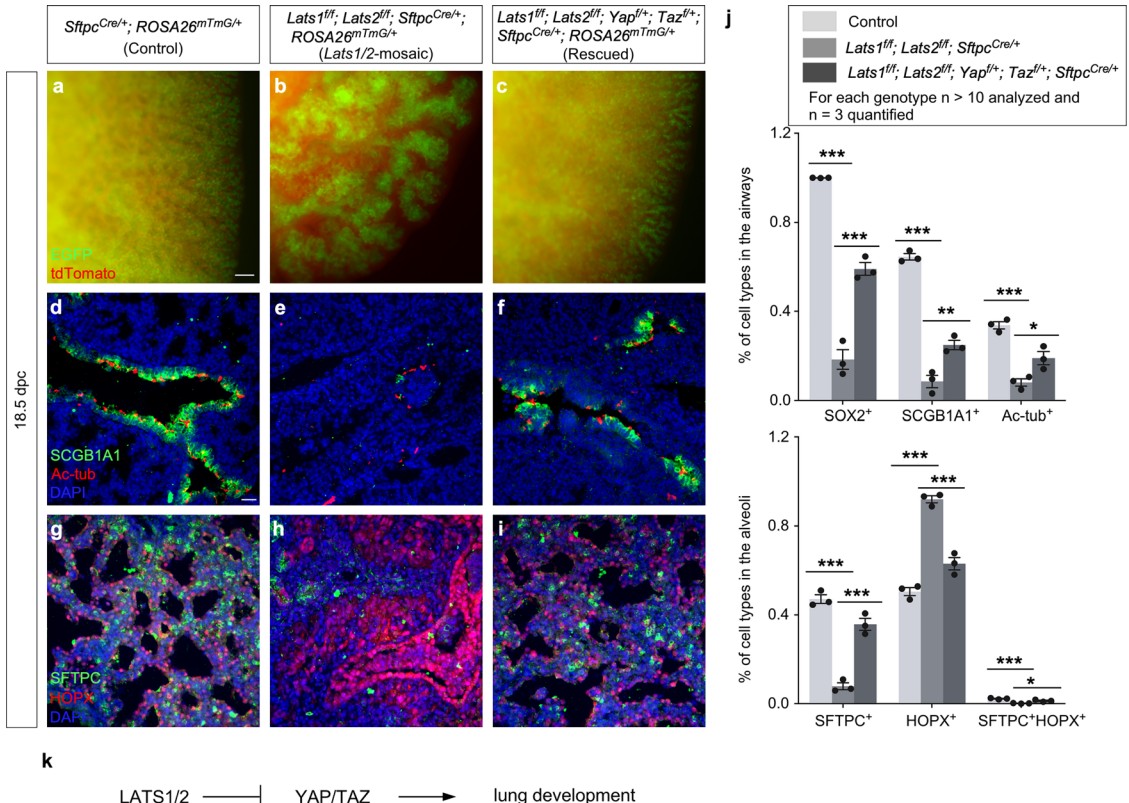

**Fig. 2 | A *Lats1/2–Yap/Taz* axis regulates lung development. a–c** Whole-mount imaging of dissected lungs from *Sftpc^{Cre/+}; ROSA26^{mTmG/+}* (Control), *Lats1^{f/f}; Lats2^{f/f}; Sftpc^{Cre/+}; ROSA26^{mTmG/+}* (*Lats1/2*-mosaic), and *Lats1^{f/f}; Lats2^{f/f}; Yap^{f/+}; Taz^{f/+}; Sftpc^{Cre/+}; ROSA26^{mTmG/+}* (Rescued) mice at 18.5 *days post-coitus* (*dpc*). The images showed a surface view of the distal part of the lung, where the edge of the lung was visible. EGFP (green) was activated from the *ROSA26^{mTmG}* allele in almost all lung epithelial cells except those in the trachea; tdTomato (red) marked all non-epithelial cells where *Sftpc^{Cre}* was not expressed. The branching network that leads to saccules at the distal end can be traced in control lungs. By contrast, the elaborate respiratory tree was replaced by a few large, thickened stalks (lined with a disorganized epithelium) in *Lats1/2*-mosaic lungs. Removal of one allele of *Yap* and *Taz* partially rescued the lung defects in the *Lats1/2*-mosaic lungs. **d–i** Immunostaining of lung sections collected from control, *Lats1/2*-mosaic, and rescued mice at 18.5 *dpc*. SCGB1A1 (CC10) (green) marked club (Clara) cells; acetylated tubulin (Ac-tub) (red) marked ciliated cells in the airways, SFTPC (green) labeled alveolar epithelial type 2 (AT2) cells, and HOPX (red) detected alveolar epithelial type 1 (AT1) cells. Partial rescue of the lung phenotypes was associated with the appearance of cell types in the conducting airways and the saccules. **j** Quantification of the percentage of SOX2⁺, SCGB1A1⁺, and Ac-tub⁺ cells in the airways and SFTPC⁺, HOPX⁺, and SFTPC⁺HOPX⁺ cells in the saccules of control, *Lats1/2*-mosaic, and rescued mice at 18.5 *dpc* (*n* = 3 for each genotype). Nuclei for SCGB1A1⁺ cells were used for cell counting. **k** Schematic diagram of a genetic pathway of Hippo signaling that controls lung development. All values are mean ± SEM. (*) $p < 0.05$; (**) $p < 0.01$; (***) $p < 0.001$ (two-way ANOVA). Scale bars, 100 µm (**a–c**), 25 µm (**d–i**).

progenitors (Fig. 2k). This information is critical for modeling lung development based on a *Lats1/2–Yap/Taz* axis.

## A fraction of SOX9⁺ tip progenitors at the distal subdomain of lung branches is sufficient to drive lung outgrowth through bifurcation

The seemingly preserved lung size in *Lats1/2* mosaic mice prompted us to investigate if it was associated with the size of the SOX9⁺ domain. *Sftpc^{Cre}* inactivated *Lats1* and *Lats2* only in a subdomain of SOX9⁺ cells, as indicated by pYAP loss (Fig. 3a–f, Supplementary Fig. 3). In regions where *Lats1/2* was removed by *Sftpc^{Cre}*, a corresponding reduction in pYAP levels was observed, suggesting YAP/TAZ activation (Fig. 3a–f, Supplementary Fig. 3). Specifically, pYAP (and thus *Lats1/2*) expression was estimated to be preserved in 15–50% of the distal SOX9⁺ subdomain where the relative pYAP intensity was also maintained (Fig. 3i, Supplementary Fig. 3). By contrast, pYAP expression disappeared in a majority of the more proximal SOX9⁺ subdomain and a fraction of the SOX2⁺ domain (Fig. 3i), which is proximal to the SOX9⁺ domain and forms the future conducting airways. Thus, *Lats1^{f/f}; Lats2^{f/f}; Sftpc^{Cre/+}* mice produced a mosaic pattern of YAP/TAZ activation in the developing lungs (Fig. 3g, h). This unique property allowed us to evaluate how regulating SOX9⁺ progenitors in a subdomain of the lung epithelium affects lung growth and development.

Selective YAP/TAZ activation in *Lats1/2*-mosaic lungs was correlated with the loss of SOX9⁺ cells in the proximal subdomain with higher YAP/TAZ activity (Fig. 3j–o). Moreover, 15–50% of the distal SOX9⁺ subdomain was retained in the mutant branches (Fig. 3p). Dichotomous branching (bifurcation) was unperturbed at the distal tips of *Lats1/2*-mosaic lungs, where *Lats1/2* expression was preserved. In addition, the rate of EdU incorporation in the most distal SOX9⁺ regions where *Lats1/2* were intact was similar to that in control lungs (Supplementary Fig. 4). The apparent preservation of the mutant lung size suggests that only a fraction of the distal SOX9⁺ cells is required for lung outgrowth. In control lungs, new buds were generated through domain branching (lateral sprouting) from the stalk (Fig. 3q, r). By contrast, in regions proximal to the distal tips where *Lats1/2* were eliminated in *Lats1/2*-mosaic lungs, lung branching through domain branching was significantly reduced (Fig. 3s–u). These regions were instead lined by an unbranched, disorganized epithelium (Fig. 3v, w). These observations suggest that lungs produced by only a fraction of the bifurcation events that are distributed throughout different branches can reach the size of control lungs. The branching defects in the mutant lungs were discernible at 13.5 *dpc* and became more pronounced at 14.5 *dpc*. The trachea and main stem bronchi were unaffected in the mutant lungs. Despite the apparent preservation of the

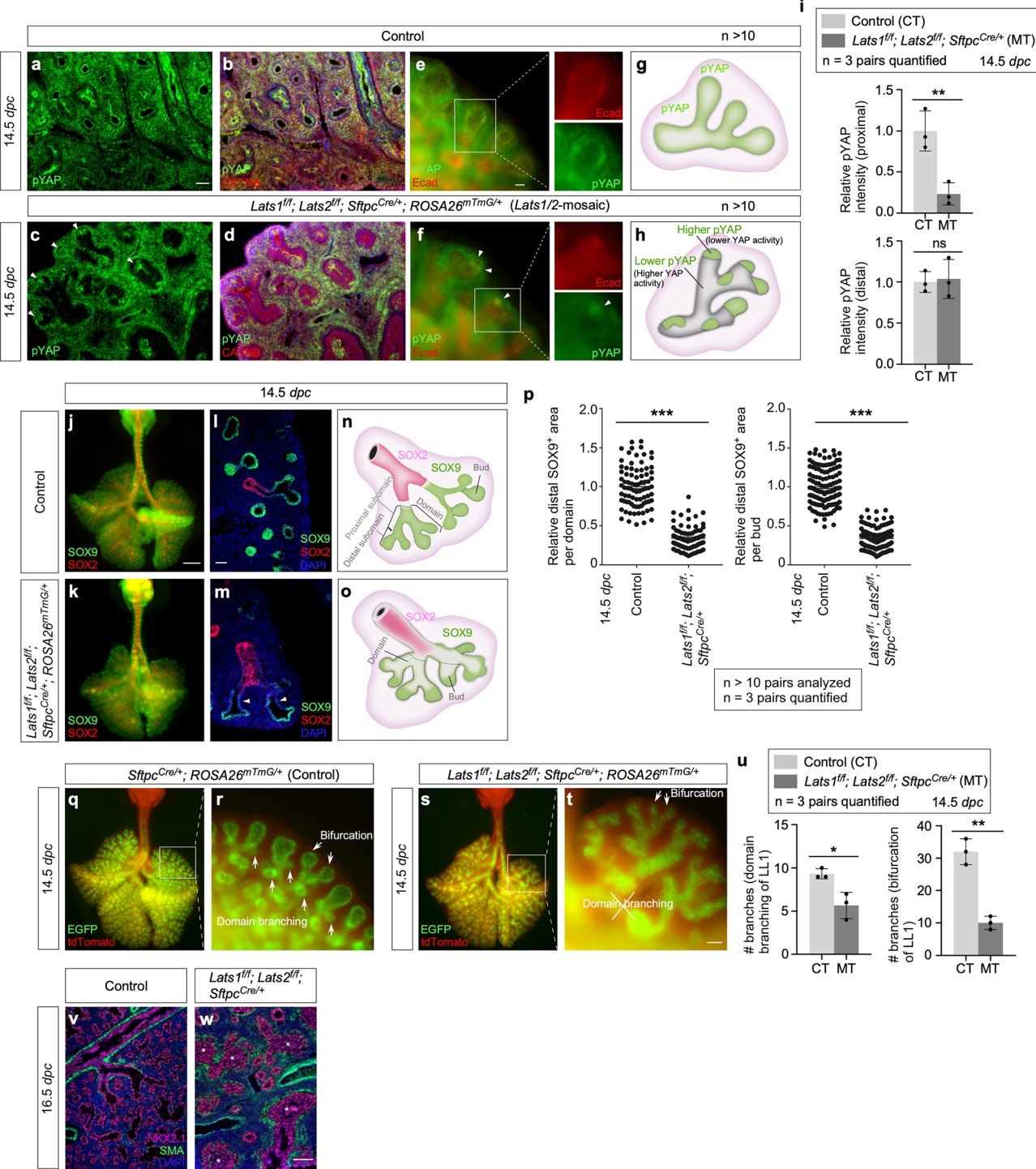

lung size, the branches located more proximally within *Lats1/2*-mosaic lungs became disorganized as development proceeded.

Together, when the non-distal SOX9+ pool was removed by *Sftpc^Cre*, lung outgrowth was largely preserved. By contrast, when the entire SOX9+ pool was removed by *Shh^Cre*, lung outgrowth was blocked. These results suggest that a SOX9+ progenitor pool at the most distal subdomain is sufficient to direct overall lung outgrowth through bifurcation.

### YAP/TAZ levels regulate cell differentiation in the transition zone (proximal to the distal tips) to produce the conducting airways in mice

The branching patterns in *Lats1^f/f; Lats2^f/f; Sftpc^Cre/+* (*Lats1/2*-mosaic) lungs appeared similar to those in control lungs up to 12.5 *dpc*

(Supplementary Fig. 5). We noticed that the SOX2+ domain in *Lats1/2* mosaic lungs gradually shrank from 13.5 to 16.5 *dpc* (Fig. 4a–d). This reduction was associated with a failure to form the conducting airways in *Lats1/2* mosaic lungs (Fig. 4e, f). These results suggest that the loss of *Lats1/2* in the SOX2+ domain abolished SOX2 expression (Fig. 4g, h). SOX9−SOX2− cells in the conducting airways of the mutant lungs then turned on markers of AT1 cells as described below.

In regions that correspond to the more proximal SOX9+ subdomain, *Lats1/2* were eliminated by *Sftpc^Cre* in the majority of this subdomain. These *Lats1/2*-deficient cells not only ceased to express SOX9 but also failed to express SOX2 (Fig. 4i), which is associated with cell-type formation in the conducting airways. Loss of SOX9 expression could be detected at 13.5 *dpc* in *Lats1/2*-mosaic lungs, and the

**Fig. 3 | YAP/TAZ activity determines the cell fates of the conducting airways.**
**a–d** Immunofluorescence of lung sections from control and *Lats1$^{f/f}$; Lats2$^{f/f}$; Sftpc$^{Cre/+}$; ROSA26$^{mTmG/+}$* (*Lats1/2*-mosaic) mice at 14.5 days post-coitus (*dpc*). *Lats1/2* removal activated YAP/TAZ as shown by the reduction of phospho-YAP (pYAP). Loss of pYAP is thus indicative of YAP/TAZ activation. pYAP (green) was detected along the proximo-distal axis of lung branches in controls. By contrast, in *Lats1/2*-mosaic lungs, pYAP was retained at the distal tips of lung branches (white arrowheads in **c**), but it was absent from the proximal part of the lung branches. CATNB (β-catenin) (red) labeled adherens junctions. **e, f** Whole-mount immunostaining of lungs from control and *Lats1$^{f/f}$; Lats2$^{f/f}$; Sftpc$^{Cre/+}$; ROSA26$^{mTmG/+}$* mice at 14.5 *dpc*. White arrowheads in (**f**) pointed to residual pYAP signals at the distal tips. The boxed areas were enlarged to visualize individual Ecad (epithelial marker) and pYAP signals. Ecad (E-cadherin) (red) labeled adherens junctions. **g, h** Schematic diagrams of pYAP distribution along the lung branches in control and *Lats1/2*-mosaic lungs. 15–50% of pYAP was estimated to be preserved in the distal SOX9$^+$ subdomain; it reflects branch-to-branch variations in *Lats1/2* loss. **i** Quantification of the relative pYAP intensity in the proximal and distal domains of lung branches in control (CT) and *Lats1/2*-mosaic (MT) lungs at 14.5 *dpc* (*n* = 3 pairs). **j, k** Whole-mount immunostaining of representative lungs (ventral view) from control and *Lats1$^{f/f}$; Lats2$^{f/f}$; Sftpc$^{Cre/+}$; ROSA26$^{mTmG/+}$* (*Lats1/2*-mosaic) mice at 14.5 *dpc*. SOX9 (green) labeled the distal airway epithelium, while SOX2 (red) labeled the proximal airway epithelium. The *Lats1/2*-mosaic lungs appeared slightly smaller but denser. No difference in proliferation was noted at 14.5 *dpc*. Together, the wet weights were similar between the control and *Lats1/2*-mosaic lungs. **l, m** Immunostaining of lung sections from control and *Lats1$^{f/f}$; Lats2$^{f/f}$; Sftpc$^{Cre/+}$; ROSA26$^{mTmG/+}$* (*Lats1/2*-mosaic) mice at 14.5 *dpc*. SOX9$^+$ signal was retained in the distal lung buds but reduced in the proximal part of lung branches (white arrowheads in (**m**)) in *Lats1/2*-mosaic mice. **n, o** Schematic diagrams of SOX9 and SOX2 distribution in control and *Lats1/2*-mosaic lungs. Note that one SOX9$^+$ domain contains multiple distal buds. One SOX9$^+$ domain contains the proximal SOX9$^+$ subdomain and the distal SOX9$^+$ subdomain. SOX9$^+$ cells are lost in the proximal SOX9$^+$ subdomain and a fraction of the distal SOX9$^+$ subdomain in *Lats1/2*-mosaic lungs. **p** Quantification of the relative SOX9$^+$ area in each distal domain or distal bud (*n* = 3 for each genotype). (**q–t** Whole-mount imaging of lungs (ventral view for (**q**) and (**s**)) from control and *Lats1$^{f/f}$; Lats2$^{f/f}$; Sftpc$^{Cre/+}$; ROSA26$^{mTmG/+}$* (*Lats1/2*-mosaic) mice at 14.5 *dpc*. Bifurcation in *Lats1/2*-mosaic lungs (white arrows in (**t**)) was preserved while domain branching was disrupted. The boxed area in Fig. 3s contains daughter branches of the L.L1 branch. Domain branching of L.L1[1] gives rise to lung branches along the stalk and epithelial buds near the surface in control lungs (see Fig. 3q, r). Figure 3s showed loss or reduced epithelial buds near the surface that are derived from the L.L1 branch. Figure 3s also exhibited loss or reduced branches along the stalk of the L.L1 branch. These results are consistent with defective domain branching of the L.L1 branch in *Lats1/2*-mosaic lungs. **u** Quantification of the number of branches through domain branching or bifurcation in the L.L1 branch of control (CT) and *Lats1/2*-mosaic (MT) lungs at 14.5 *dpc* (*n* = 3 pairs). **v, w** Immunofluorescence of lung sections from control and *Lats1$^{f/f}$; Lats2$^{f/f}$; Sftpc$^{Cre/+}$* (*Lats1/2*-mosaic) mice at 16.5 *dpc*. NKX2.1 (magenta) marked lung epithelial cells, while SMA (green) labeled mesenchymal myofibroblasts. The disorganized epithelium (asterisks) was observed in *Lats1/2*-mosaic lungs, which was also revealed in Fig. 1k and Fig. 2h. All values are mean ± SEM. (*) *p* < 0.05; (**) *p* < 0.01; (***) *p* < 0.001; ns, not significant (two-tailed Student's *t*-test). Scale bars, 50 µm (**a–d**), 25 µm (**e, f**), 0.5 mm (**j, k, q, s**), 50 µm (**l, m**), 100 µm (**r, t**), 50 µm (**v, w**).

---

number of SOX9$^-$SOX2$^-$ cells, which occupied the intermediate domain between the SOX9$^+$ and SOX2$^+$ domains, significantly increased from 13.5 to 16.5 *dpc* (Fig. 4i, j). At 16.5 *dpc*, SOX9$^-$SOX2$^-$ cells were distributed in a mosaic pattern in the entire lung except the most distal part, where *Sftpc$^{Cre}$* failed to eliminate *Lats1/2* (Fig. 4g, h). This suggests that *Lats1* and *Lats2* were inactivated in the progeny produced from the distal SOX9$^+$ progenitors as the descendants took up a more proximal location. SOX9$^-$SOX2$^-$ cells can originate from (1) SOX9$^+$ cells that lost SOX9 and failed to turn on SOX2; and (2) SOX2$^+$ cells that turned off SOX2 in *Lats1/2*-deficient epithelium. The size of the residual SOX9$^+$ domain in *Lats1/2*-mosaic lungs ranged from 15–50% of that in control lungs, as described above.

To gain molecular insight into how *Lats1/2* regulate lung branching, we collected lungs from control and *Lats1/2*-mosaic mice at 13.5 *dpc* and performed bulk RNA-seq analysis to probe the changes in transcriptomes in the absence of *Lats1/2* (Fig. 4k). The expression levels of canonical YAP targets, such as *Ctgf* (connective tissue growth factor, Ccn2), *Cyr61* (cysteine-rich protein 61, CCN1), and *Ajuba* (ajuba LIM protein), were increased in *Lats1/2*-mosaic lungs compared to controls (Fig. 4k). This was confirmed by qPCR analysis (Fig. 4l). These results indicate that YAP targets in the lung epithelium are activated in the absence of *Lats1/2*. Consistent with a reduced SOX9$^+$ domain at the distal tip of *Lats1/2*-mosaic lungs, bulk RNA-seq data revealed reduced expression of markers of the distal domain, such as *Sox9*, *Id2*, *Etv5*, and *Wnt7b* (Fig. 4k). Several regulators of the cell cycle (*e.g.*, *Cdkn2b*) or growth-promoting pathways (*e.g.*, *Wnt*) were upregulated in the mutant lungs (Fig. 4m).

The stalks populated by SOX9$^-$SOX2$^-$ cells in *Lats1/2*-mosaic lungs failed to undergo domain branching and presented as thickened layers of epithelial cells (Fig. 4f). Consistent with this phenotype, pathway analysis of bulk RNA-seq revealed upregulation of pathways that regulate cell proliferation, focal adhesion, the actin cytoskeleton, cell migration, cell motility, and ECM-receptor interaction in the mutant lungs (Fig. 4k, m, Supplementary Fig. 6). Disruption of these pathways is expected to perturb proper cell-cell interactions required for branching.

The rate of EdU incorporation in SOX9$^-$SOX2$^-$ cells in *Lats1/2*-mosaic lungs was similar to or slightly higher than that in control lungs (Supplementary Fig. 4). This result suggests that the failure of domain branching caused by YAP/TAZ activation was not due to reduced cell proliferation. Together, these results reveal the fate of SOX9$^+$ cells in response to different levels of YAP/TAZ activity during lung branching.

## YAP/TAZ activation in SOX9$^+$ progenitors promotes the AT1 fate, while YAP/TAZ inactivation in SOX9$^+$ progenitors increases the AT2-to-AT1 ratio in the mouse lungs

*Sftpc$^{Cre}$* is active prior to the differentiation of SOX9$^+$ progenitors to produce alveolar epithelial cells and does not provide the genetic setting to assess the developmental potential of SOX9$^+$ progenitors during sacculation. To address this issue, we deployed *Sox9$^{CreER}$* [28] to remove *Lats1/2* in SOX9$^+$ progenitors at the distal lung epithelium. We generated control and *Lats1$^{f/f}$; Lats2$^{f/f}$; Sox9$^{CreER/+}$* mice and administered tamoxifen to pregnant females at 14.5 or 15.5-16.0 *dpc* (Fig. 5a, b). Analysis of mouse lungs from *Lats1$^{f/f}$; Lats2$^{f/f}$; Sox9$^{CreER/+}$* mice collected at 18.5 *dpc* revealed an increased ratio of AT1-to-AT2 compared to control lungs (Fig. 5c–h). Fewer AT2 (SFTPC$^+$) cells were detected in the mutant lungs (Fig. 5d, f). Thus, higher YAP/TAZ levels in SOX9$^+$ progenitors drive the transition to the AT1 fate.

To further test our model, we used *Sox9$^{CreER}$* to remove *Yap/Taz* in SOX9$^+$ progenitors in the distal lung epithelium. We generated control and *Yap$^{f/f}$; Taz$^{f/f}$; Sox9$^{CreER/+}$* mice and administered tamoxifen to pregnant females at 13.5 or 15.5–16.0 *dpc* (Fig. 5i, j). Analysis of mouse lungs from *Yap$^{f/f}$; Taz$^{f/f}$; Sox9$^{CreER/+}$* mice collected at 18.5 *dpc* revealed an increased ratio of AT2-to-AT1 compared to control lungs (Fig. 5k–p). Moreover, aggregates of AT2 cells were observed in the mutant lungs (Fig. 5l, n). These findings are consistent with our model in which lower YAP/TAZ levels maintain the progenitor pool.

To perturb Hippo signaling in a subset of SOX9$^+$ progenitors, we generated control and *Yap$^{f/f}$; Taz$^{f/f}$; Sftpc$^{CreER/+}$* lungs and administered tamoxifen to these animals at 13.5 *dpc* (Fig. 5q). This setup enabled the manipulation of Hippo signaling by *Sftpc$^{CreER}$* [29] in SFTPC$^+$ cells. Likewise, *Yap$^{f/f}$; Taz$^{f/f}$; Sftpc$^{CreER/+}$* lungs collected at 18.5 *dpc* exhibited an increased ratio of AT2-to-AT1 compared to control lungs, as well as AT2 aggregation (Fig. 5r-t). Together, these studies support a critical role of YAP/TAZ levels in controlling the AT1/AT2 fate.

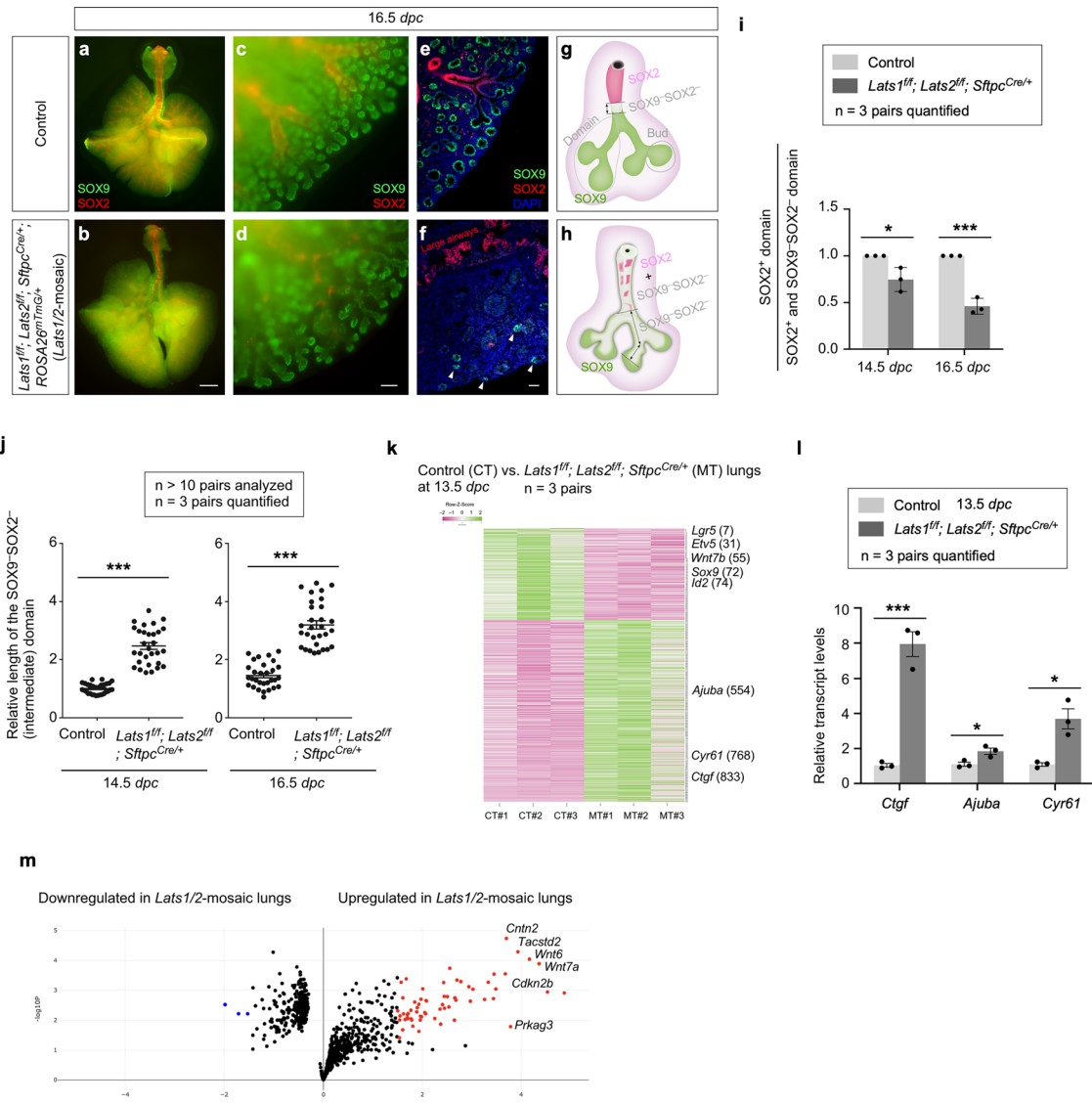

**Fig. 4 | YAP/TAZ activity regulates the SOX9-to-SOX2 transition. a–d** Whole-mount immunostaining of dissected lungs (ventral view for (**a**) and (**b**)) from control and *Lats1^f/f^; Lats2^f/f^; Sftpc^Cre/+^;ROSA26^mTmG/+^* (*Lats1/2*-mosaic) mice at 16.5 *days post-coitus* (*dpc*). SOX9 (green) and SOX2 (red) marked the distal and proximal airway epithelium, respectively. A reduction in SOX9 and SOX2 signals was discerned in the more proximal regions, while the SOX9 signal was preserved in the distal tips of *Lats1/2*-mosaic lungs. **e**, **f** Immunostaining of lung sections from control and *Lats1/2*-mosaic mice at 16.5 *dpc*. In control lungs, the SOX9^-^SOX2^-^ region sandwiched between the SOX9^+^ and SOX2^+^ domains became more noticeable than in earlier stages. In the absence of *Lats1/2*, the SOX9^-^SOX2^-^ region was significantly expanded. White arrowheads in (**f**) pointed to residual distal SOX9^+^ cells. **g**, **h** Schematic diagrams depicting the distribution of the SOX9^+^, SOX9^-^SOX2^-^ and SOX2^+^ domains in control and *Lats1/2*-mosaic lungs. **i** Quantification of the SOX2^+^ domain within the domain that contains both SOX2^+^ and SOX9^-^SOX2^-^ cells in control and *Lats1/2*-mosaic lungs at 14.5 and 16.5 *dpc* (*n* = 3 pairs). Areas for each domain were measured in ImageJ. **j** Quantification of the length of the SOX9^-^SOX2^-^ domain in control and *Lats1/2*-mosaic lungs at 14.5 and 16.5 *dpc* (*n* = 3 pairs). The relative length refers to the relative length between

control and *Lats1/2*-mosaic lungs measured in ImageJ. **k** Heatmap of gene expression by bulk RNA-seq analysis from control (CT) and *Lats1/2*-mosaic (MT) lungs at 13.5 *dpc*. Out of the 919 genes with differential expression, 308 genes were downregulated, and 611 genes were upregulated in *Lats1/2*-mosaic lungs. The number next to the gene indicates its position in the heatmap. 1–308 are genes downregulated, and 309–919 are genes upregulated in *Lats1/2*-mosaic lungs. Canonical YAP targets, such as *Ctgf*, *Ajuba*, and *Cyr61*, were upregulated in *Lats1/2*-mosaic lungs. **l** qPCR analysis of control and *Lats1/2*-mosaic lungs at 13.5 *dpc* (*n* = 3 pairs). The transcript levels of canonical YAP targets, such as *Ctgf*, *Ajuba*, and *Cyr61*, were upregulated in *Lats1/2*-mosaic lungs. **m** Volcano plots of up-regulated and down-regulated genes in *Lats1/2*-mosaic lungs at 13.5 *dpc*. Several regulators of the cell cycle (*e.g.*, *Cdkn2b*) or growth-promoting pathways (*e.g.*, *Wnt*) were upregulated in *Lats1/2*-mosaic lungs. The blue dots represent the most down-regulated genes with expression levels less than 3.5 fold, while the red dots represent the most up-regulated genes with expression levels greater than 3.5 fold. All values are mean ± SEM. (*) $p < 0.05$; (***) $p < 0.001$ (two-tailed Student's *t*-test). Scale bars, 1 mm (**a**, **b**), 100 μm (**c**, **d**), 100 μm (**e**, **f**).

## scRNA-seq and multiomics analysis of control and *Lats1/2*-mosaic lungs reveals different transitional states and candidate regulators during branching

To uncover the molecular mechanisms by which YAP/TAZ control the SOX9^+^ progenitor pool, we performed scRNA-seq analysis on cells isolated from control (*n* = 13) and *Lats1^f/f^; Lats2^f/f^; Sftpc^Cre/+^* (*Lats1/2*-

mosaic) (*n* = 13) mouse lungs at 14.5 *dpc*. Lung cells were dissociated through enzymatic digestion and purified against immune cells (CD45) and hematopoietic cells (TER119) by fluorescence-activated cell sorting (FACS). We performed scRNA-seq analysis using the 10x Genomics platform. Analysis of scRNA-seq data was conducted using Seurat, Monocle, and other bioinformatics programs. A total of 13,985 control

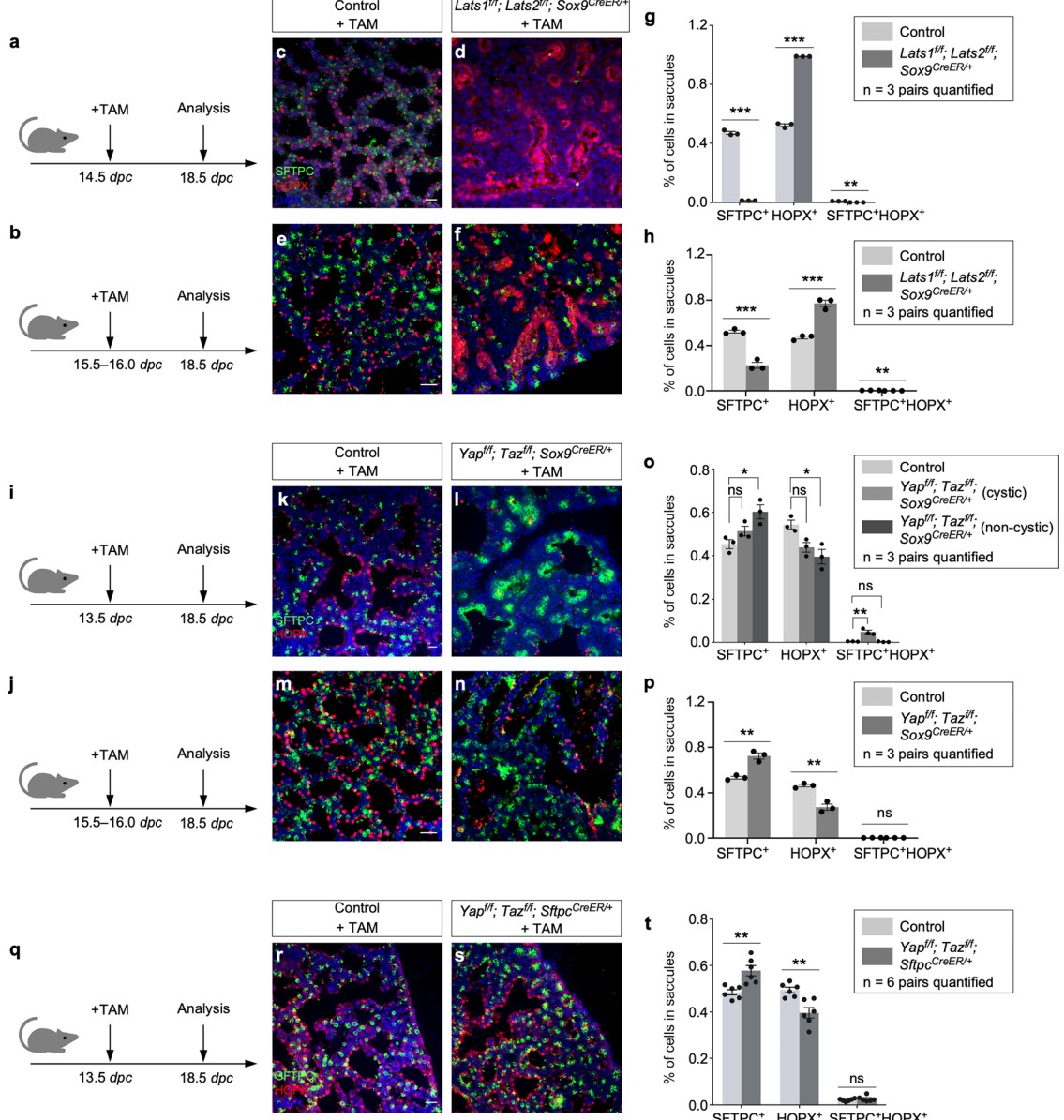

**Fig. 5 | YAP/TAZ activity controls the AT1–AT2 fate. a, b** Schematic diagram of the time course of tamoxifen (TAM) administration and lung collection for control and *Lats1^f/f^; Lats2^f/f^; Sox9^CreER/+^* (Mutant) mice. *dpc, days post-coitus.* **c–f** Immunofluorescence of representative areas in control and *Lats1^f/f^; Lats2^f/f^; Sox9^CreER/+^* lungs. AT1 (HOPX^+^) (red) cells were prevalent in the mutant lungs, whereas AT2 (SFTPC^+^) (green) cells were scant. **g, h** Quantification of the number of SFTPC^+^ (AT2) cells, HOPX^+^ (AT1) cells, and SFTPC^+^ HOPX^+^ cells in control and *Lats1^f/f^; Lats2^f/f^; Sox9^CreER/+^* lungs collected at 18.5 *dpc* (*n* = 3 pairs). Loss of *Lats1/2* led to an increased ratio of AT1 to AT2 cells. **i, j** Schematic diagram of the time course of tamoxifen administration and lung collection for control and *Yap^f/f^; Taz^f/f^; Sox9^CreER/+^* mice. **k–n** Immunofluorescence of representative areas in control and *Yap^f/f^; Taz^f/f^; Sox9^CreER/+^* lungs. AT2 cells were prevalent in the non-cystic region. The cystic wall

was lined primarily by AT1 cells. **o, p** Quantification of the number of SFTPC^+^ (AT2) cells, HOPX^+^ (AT1) cells, and SFTPC^+^HOPX^+^ cells in control and *Yap^f/f^; Taz^f/f^; Sox9^CreER/+^* lungs collected at 18.5 *dpc* (*n* = 3 pairs). Loss of *Yap/Taz* led to an increased ratio of AT2 to AT1 cells. **q** Schematic diagram of the time course of tamoxifen administration and lung collection for control and *Yap^f/f^; Taz^f/f^; Sftpc^CreER/+^* mice. **r, s** Immunofluorescence of representative areas in control and *Yap^f/f^; Taz^f/f^; Sftpc^CreER/+^* lungs. **t** Quantification of the number of SFTPC^+^ (AT2) cells, HOPX^+^ (AT1) cells, and SFTPC^+^HOPX^+^ cells in control and *Yap^f/f^; Taz^f/f^; Sftpc^CreER/+^* lungs collected at 18.5 *dpc* (*n* = 6 pairs). Likewise, the loss of *Yap/Taz* led to an increased ratio of AT2 to AT1 cells. All values are mean ± SEM. (*) $p < 0.05$; (**) $p < 0.01$; (***) $p < 0.001$; ns, not significant (two-tailed Student's *t*-test). Scale bars, 25 μm (**c, d, r, s**), 25 μm (**e, f, m, n**), 25 μm (**k, l**).

and 13,275 mutant lung cells were analyzed. We also re-analyzed published scRNA-seq of control lungs at 12.5 and 14.5 *dpc*[30–32] for comparative studies (Supplementary Figs. 7, 8).

In control lungs at 14.5 *dpc*, several clusters were discerned among *Epcam^+^* epithelial cells (Fig. 6a, b, Supplementary Fig. 9a). We computationally removed non-epithelial cells for cluster analysis. Based on gene expression, we inferred that cells in cluster 0 (*Sox9^+^*) went through cluster 1 (a transitional state) to reach clusters 2 and 3 (*Sox2^+^*)

(Fig. 6c). This trajectory could be in parallel (0–1–2 or 0–1–3) or in sequence (0–1–2–3). In this trajectory, the number of *Sox9^+^* cells decreased while the number of *Sox2^+^* cells increased (Supplementary Fig. 9a). Cluster 4 is a distinct cluster of *Sox2^+^* cells, which express neuroendocrine markers (see below). Expression of *Lgals3* (a marker for the reported pre-AT1 transitional state, PATS, following lung injury) and *F3* was low in control lungs at 14.5 *dpc* and in several published datasets[30,31] (Supplementary Fig. 7).

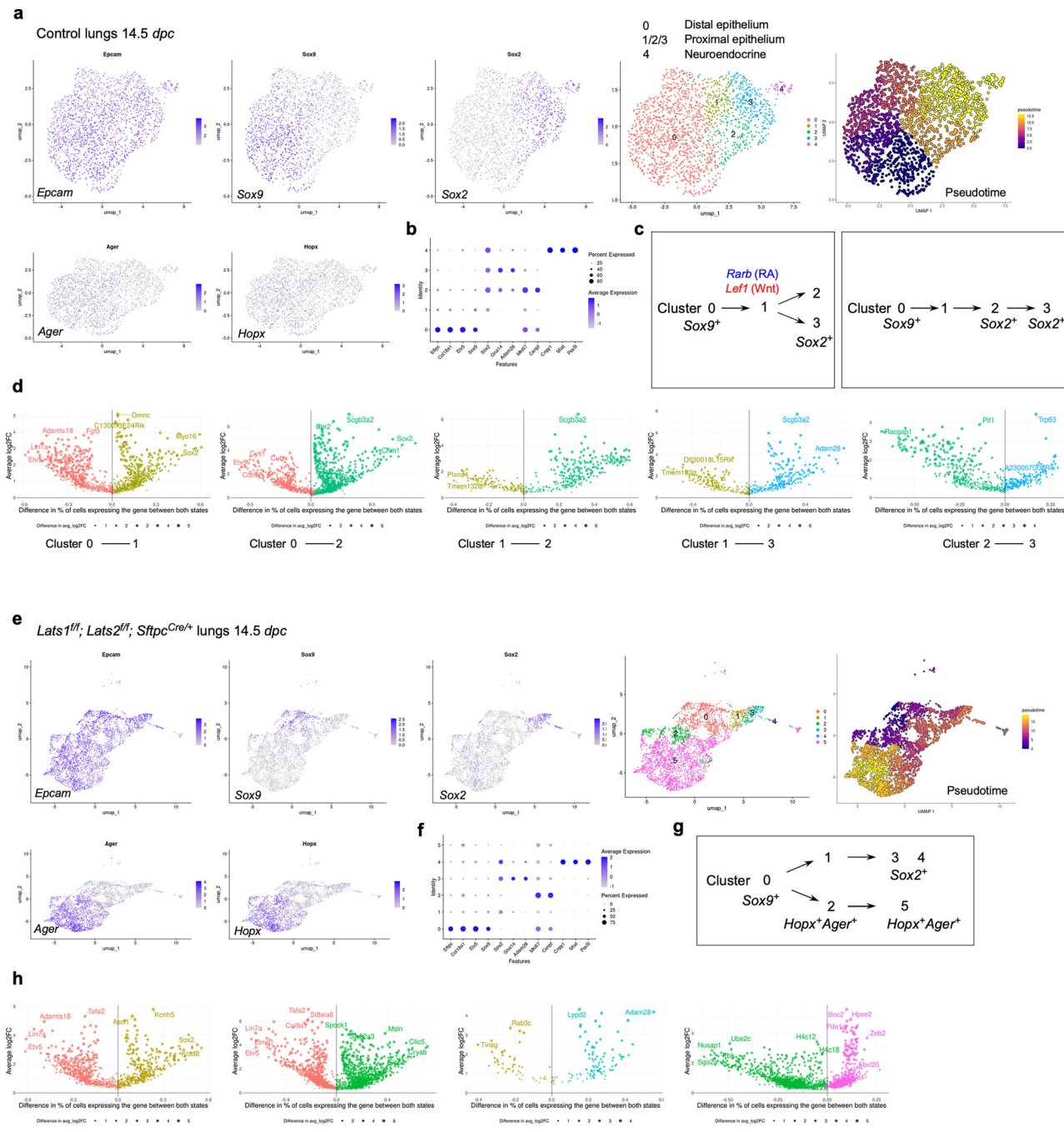

In *Lats1/2*-mosaic mouse lungs at 14.5 *dpc*, scRNA-seq analysis revealed a reduction in *Sox9*+ and *Sox2*+ cells, while the number of *Sox9*−*Sox2*− cells increased (Fig. 6e, f), consistent with our phenotypic analysis. This observation suggests that the *Sox9*+ pool was downsized, while its progeny failed to turn on *Sox2* in the absence of *Lats1/2*. Compared to the control lungs, the trajectory of cell clusters in *Lats1/2*-deficient lungs was altered (Fig. 6g). A trajectory of 0–1–3 was still present in the mutant lungs, in which cells in cluster 0 (*Sox9*+) went through cluster 1 (a transitional state) to reach cluster 3 (*Sox2*+). However, a new trajectory of 0–2–5 in the mutant lungs led to the expression of AT1 markers, such as *Hopx* and *Ager*, in clusters 2 and 5, which were also *Sox9*−*Sox2*−. This suggests that increased YAP/TAZ activity promotes the expression of AT1 markers. *Lgals3* was expressed in clusters 2 and 5, supporting the notion that these clusters represent the pre-AT1 state. We noticed that cluster 2 in the mutant lungs shared certain patterns of gene expression (heatmap) to cluster 2 in the control lungs but has lost *Sox2* expression (Supplementary Fig. 9b). Cluster 2 in the control and mutant lungs can also be distinguished by unique sets of gene expression (Supplementary Fig. 9b). Cluster 5 can be subclustered; each subcluster requires additional analysis. These studies provide key insights into how YAP/TAZ activity levels direct the fate of *Sox9*+ cells through different paths of transitional states.

We also computationally distinguished between control and mutant cells in scRNA-seq data from *Lats1/2*-mosaic lungs (Supplementary Fig. 10). Consistent with this finding, mutant cells expressed high levels of YAP targets (such as *Ctgf*, *Ajuba*, and *Amotl2*) (Supplementary Fig. 10). Mutant cells constituted the new clusters and transitional states that are only present in *Lats1/2*-mosaic lungs (Supplementary Fig. 10). This observation also suggests that the

**Fig. 6 | Multiple transitional states mediate the SOX9–to–SOX2 transition to generate the conducting airways. a** UMAP (uniform manifold approximation and projection) visualization of major cell clusters from control lungs at 14.5 days post-coitus (dpc) profiled by scRNA-seq. Only lung epithelial cells (*Epcam*⁺) were shown. Cells that expressed the featured genes were indicated. Pseudotime analysis was used to map the trajectories of cell clusters. The color gradient, ranging from dark blue to light yellow, indicates the progression of cell states over pseudotime. **b** Dot plot of a partial list of markers that were used to identify cell clusters in this study. **c** Schematic diagrams of the trajectories of cell clusters from the SOX9⁺ to SOX2⁺ states in control lungs at 14.5 dpc. The blue genes represent those that are upregulated as cells transition from the left cluster to the right in the diagram. The red genes are those that are downregulated as cells transition from the left cluster to the right in the diagram. Additional differentially expressed genes (DEGs) between different cell clusters at 14.5 dpc are listed in Supplementary Fig. 11. **d** Volcano plots of DEGs between different cell clusters in control lungs at 14.5 dpc. DEGs in a given cluster carry the same color as shown in (**a**). The y-axis represents the average change in gene expression for a given gene between the two designated clusters. The x-axis represents the difference in the percentage of cells expressing the given

gene between the two clusters shown. Changes in gene expression are statistically significant ($p < 0.05$) for all depicted DEGs (two-tailed Student's t-test). **e** UMAP visualization of major cell clusters from the lungs of *Lats1*^f/f^; *Lats2*^f/f^; *Sftpc*^Cre/+^ (*Lats1/2*-mosaic) mice at 14.5 dpc profiled by scRNA-seq. Only lung epithelial cells (*Epcam*⁺) were shown. Cells that expressed the featured genes were indicated. Pseudotime analysis was included to map the trajectories of cell clusters. The color gradient, ranging from dark blue to light yellow, indicates the progression of cell states over pseudotime. **f** Dot plot of a partial list of markers that were used to identify cell clusters in this study. **g** Schematic diagrams of the trajectories of cell clusters in *Lats1/2*-mosaic lungs at 14.5 dpc. The trajectory of SOX9⁺ to SOX2⁺ transition was present in *Lats1/2*-mosaic lungs. A new trajectory of SOX9⁺ to HOPX⁺AGER⁺ transition was found in *Lats1/2*-mosaic lungs. **h** Volcano plots of DEGs between different cell clusters in *Lats1/2*-mosaic lungs at 14.5 dpc. DEGs in a given cluster carry the same color as shown in (**e**). The y-axis represents the average change in gene expression for a given gene between the two designated clusters. The x-axis represents the difference in the percentage of cells expressing the given gene between the two clusters shown. Changes in gene expression are statistically significant ($p < 0.05$) for all depicted DEGs (two-tailed Student's t-test).

---

phenotypic consequences are due to the cell-autonomous behavior of mutant cells.

scRNA-seq analysis revealed differentially expressed genes (DEGs) in each cluster of control and *Lats1/2*-mosaic lungs at 14.5 dpc (Fig. 6d, h). These DEGs not only define the cell clusters but also can be used to validate the path from the *Sox9*⁺ to the *Sox2*⁺ state. DEGs associated with the transition between cell clusters are also candidates for fate change. In particular, we identified transcription factors among DEGs (Supplementary Fig. 11), and a transcription factor network can be constructed (Supplementary Fig. 12). Several of these transcription factors have been previously reported to regulate lung branching and/or cell differentiation. For instance, retinoic acid receptor β (RARB), a nuclear receptor and ligand-activated transcription factor, was enriched when cluster 0 transitions to 1 or 2 at 14.5 dpc (Supplementary Fig. 11). Retinoic signaling has been implicated in lung branching[33]. Our scRNA-seq analysis has identified a candidate cell population in which RA signaling controls early lung development. The transition of cluster 0 to 1 or 2 was also associated with the downregulation of *Lef1*, a transducer of Wnt signaling, and *Rbpjl*, an antagonist of Notch signaling. Genetic loss of *Lef1* and its paralogs in mice accelerated differentiation into SOX2 progeny[34]. Thus, our scRNA-seq has identified a potential target population of Wnt and Notch signaling, the downregulation of which is required for state transition. Several other DEGs have been implicated in lung development or cell fate specification (e.g., *Etv5, Sp5, Six1*)[35–37]. Our analysis thus provides a cellular framework for how these regulators function. Moreover, several other regulators and transcription factors are not known to control lung branching and/or cell differentiation (e.g., *Npas2*, which regulates circadian rhythm). Our scRNA-seq analysis could reveal their new functions in the developing lung.

To reveal the signaling network that controls the SOX9⁺ progenitors, we performed multiomics analysis on cells isolated from control (n = 12) and *Lats1*^f/f^; *Lats2*^f/f^; *Sftpc*^Cre/+^ (*Lats1/2*-mosaic) (n = 11) lungs at 14.5 dpc. We isolated nuclei from control and *Lats1/2*-mosaic lung cells at 14.5 dpc and performed multiomics analysis using the 10x Genomics platform. snRNA-seq and snATAC-seq data from 2339 control and 3292 mutant lung nuclei were analyzed.

We identified differentially accessible regions (DARs) between SOX9⁺ and SOX2⁺ cells in control lungs at 14.5 dpc (Supplementary Fig. 13). Multiple motifs within DARs were revealed. The motifs enriched in either SOX9⁺ or SOX2⁺ cells provide insight into potential regulators for differentiating SOX9⁺ cells into SOX2⁺ cells in the conducting airways. Several DEGs associated with the transition between the SOX9⁺ and SOX2⁺ states contain the TEAD motif in their regulatory

regions (Supplementary Fig. 13). This suggests that YAP/TAZ utilizes a regulatory network to control the transition of SOX9⁺ to SOX2⁺ cells.

Together, we have identified the developmental path of SOX9⁺ cells to produce SOX2⁺ cells and how their transitional states are affected by YAP/TAZ levels and regulators.

## scRNA-seq and multiomics analysis of control and *Lats1/2*–mosaic lungs uncovers different transitional states associated with the progression of the AT1 fate and candidate regulators during sacculation

To unveil the molecular mechanisms by which YAP/TAZ controls the SOX9⁺ progenitor pool during sacculation, we performed scRNA-seq on cells isolated from control (n = 4) and *Lats1*^f/f^; *Lats2*^f/f^; *Sftpc*^Cre/+^ (*Lats1/2*-mosaic) (n = 4) mouse lungs at 17.5 dpc. Lung cell isolation and scRNA-seq were performed as described above. A total of 8979 control and 31,392 mutant lung cells were analyzed. Likewise, we re-analyzed published scRNA-seq of control lungs at 15.0, 15.5, 16.5, 17.5, and 18.5 dpc[30,31,34,38,39] for comparative studies (Supplementary Figs. 14, 15).

In control lungs at 17.5 dpc, several cell clusters were discerned among *Epcam*⁺ epithelial cells (Fig. 7a, b, Supplementary Fig. 16). Based on gene expression, we inferred that cells in cluster 0 (*Sox9*⁺*Sftpc*⁺) went through cluster 1 (*Sftpc*⁺*Hopx*⁺*Ager*⁺) to reach clusters 2 and 3 (*Hopx*⁺*Ager*⁺) (Fig. 7c). In this trajectory, *Sox9*⁺ cells produced AT1/AT2 cells and then AT1 cells. *Sox9*⁻*Sftpc*⁺ cells in cluster 0 lost *Sox9* expression and likely generated AT2 cells. *Hopx*⁺*Ager*⁺ cells in clusters 2 and 3 expressed *Lgals3*, suggesting a more mature state compared to 14.5 dpc.

We found that the cluster of *Hopx*⁺*Ager*⁺ (AT1) cells was significantly larger in *Lats1/2*-mosaic lungs (Fig. 7e, f). By contrast, the cell cluster that expresses AT2 markers was greatly diminished in *Lats1*^f/f^; *Lats2*^f/f^; *Sftpc*^Cre/+^ mouse lungs (Fig. 7e). This supports our model in which activation of YAP/TAZ depletes the AT2 pool and promotes the AT1 fate. The mutant *Hopx*⁺*Ager*⁺ cluster contained several subclusters, among which cluster 5 expressed *Krt8*, *Cldn4*, *Sfn*, and *Lgals3* (Fig. 7g, h).

Moreover, we performed a comparative analysis between (1) *Lats1/2*-deficient *Hopx*⁺*Ager*⁺ cells at 14.5 dpc [designated as state 1], (2) control *Hopx*⁺*Ager*⁺ cells at 17.5 dpc [state 2], (3) *Lats1/2*-deficient *Hopx*⁺*Ager*⁺ cells at 17.5 dpc [state 3], and (4) control *Hopx*⁺*Ager*⁺ cells at neonatal and adult lungs [state 4]. We found that they represent different stages of AT1 maturation (Fig. 7i, j). For instance, only states 2 and 3 (not 1) expressed *Lgals3* (which is also expressed in adult AT1), *Emp2*, and *Clic5*. Only state 3 (not 1 or 2) expressed *S100a6*, *Krt7*, and *Cryab* (which are all expressed in neonatal or adult AT1 [state 4]). These

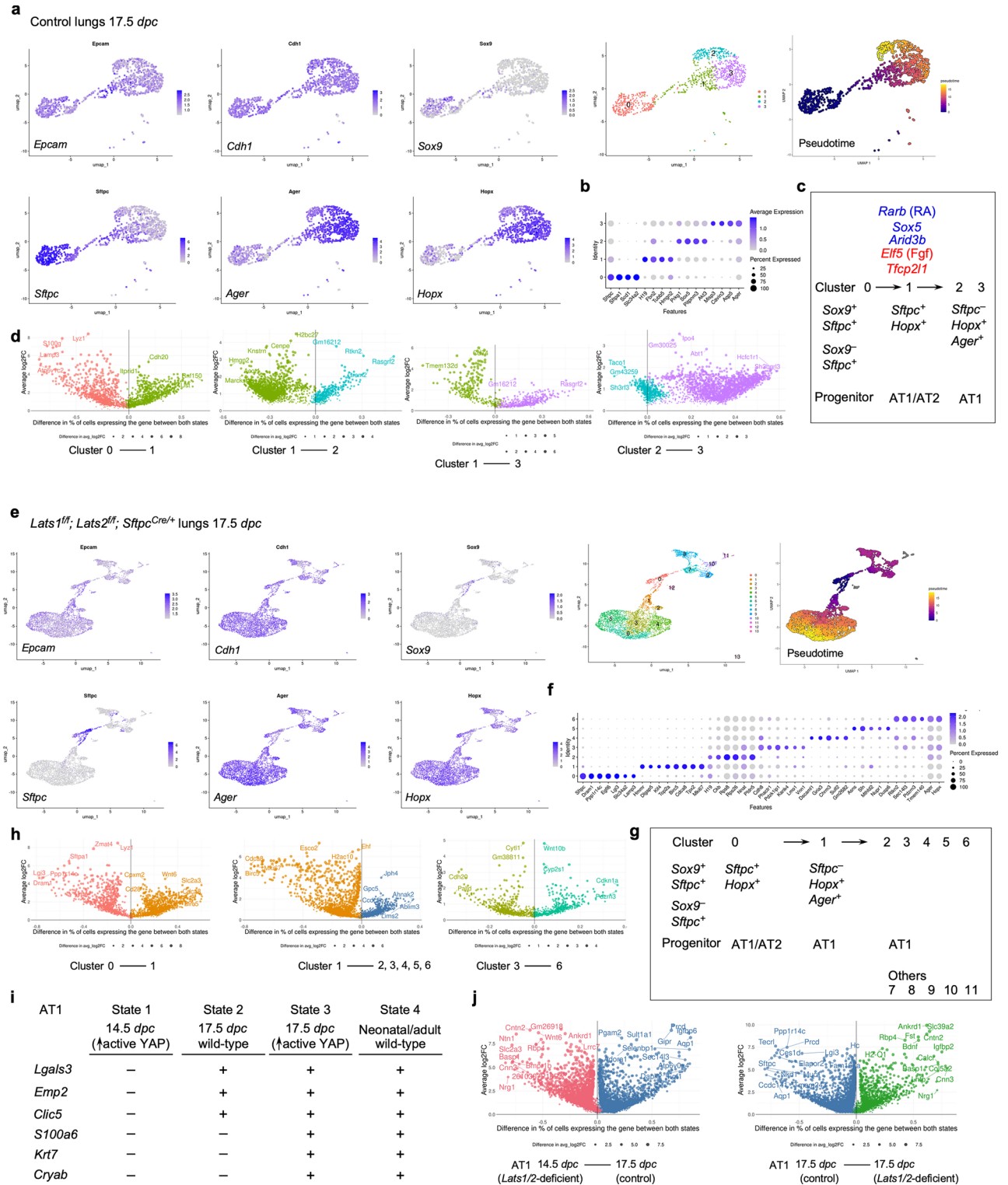

states (1–2–3) thus represent the sequential maturation of AT1 cells during development.

We also computationally distinguished between control and mutant cells in scRNA-seq data from *Lats1/2*-mosaic lungs (Supplementary Fig. 10). Consistent with this finding, mutant cells also expressed high levels of YAP targets (such as *Ctgf*, *Ajuba*, and *Amotl2*) (Supplementary Fig. 10). Mutant cells constituted the new clusters that are only present in the mutant lungs (Supplementary Fig. 10).

scRNA-seq analysis revealed differentially expressed genes (DEGs), including transcription factors among DEGs (Supplementary

Fig. 17) in each cluster of control and *Lats1/2*-mosaic lungs (Fig. 7d, h, j), and a transcription factor network can be constructed (Supplementary Fig. 18). Likewise, they will define the cell clusters and provide candidates that control fate change. For example, the expression of *Rarb* and *Sox5* was increased when cluster 0 transitioned to 1 at 17.5 *dpc*. Lung development failed to progress to the saccular stage in *Sox5* knockout mice[40], although the molecular basis of the phenotype is unknown. Our scRNA-seq analysis revealed where and when *Rarb* and *Sox5* may act. The transition from cluster 0 to 1 was associated with reduced expression of several regulators, including *Cebpa*,

**Fig. 7 | Multiple transitional states mediate the fate transition from SOX9⁺ progenitors to the alveolar epithelial cells. a** UMAP (uniform manifold approximation and projection) visualization of major cell clusters from control lungs at 17.5 *days post-coitus* (*dpc*) profiled by scRNA-seq. Only lung epithelial cells (*Epcam*⁺) were shown. Cells that expressed the featured genes were indicated. Pseudotime analysis was used to map the trajectories of cell clusters. The color gradient, ranging from dark blue to light yellow, indicates the progression of cell states over pseudotime. **b** Dot plot of a partial list of markers that were used to identify cell clusters in this study. **c** Schematic diagrams of the trajectories of cell clusters from the SOX9⁺ to HOPX⁺AGER⁺ (AT1) transition in control lungs at 17.5 *dpc*. The blue genes represent those that are upregulated as cells transition from the left cluster to the right in the diagram. The red genes are those that are downregulated as cells transition from the left to the right cluster in the diagram. Additional differentially expressed genes (DEGs) between different cell clusters at 17.5 *dpc* are listed in Supplementary Fig. 17. **d** Volcano plots of DEGs between different cell clusters in control lungs at 17.5 *dpc*. DEGs in a given cluster carry the same color as shown in (**a**). The y-axis represents the average change in gene expression for a given gene between the two designated clusters. The x-axis represents the difference in the percentage of cells expressing the given gene between the two clusters shown. Changes in gene expression are statistically significant ($p < 0.05$) for all depicted DEGs (two-tailed Student's *t*-test). **e** UMAP visualization of major cell clusters from lungs from *Lats1^f/f^; Lats2^f/f^; Sftpc^Cre/+^* (*Lats1/2*-mosaic) mice at 17.5 *dpc* profiled by scRNA-seq. Only lung epithelial cells (*Epcam*⁺) were shown. Cells that expressed the featured genes were indicated. Pseudotime analysis was included to map the trajectories of cell clusters. The color gradient, ranging from dark blue to light yellow, indicates the progression of cell states over pseudotime. **f** Dot plot of a partial list of markers that were used to identify cell clusters in this study. **g** Schematic diagrams of the trajectories of cell clusters from the SOX9⁺ to HOPX⁺AGER⁺ (AT1) transition in *Lats1/2*-mosaic lungs at 17.5 *dpc*. Additional clusters in the AT1 states were uncovered in *Lats1/2*-mosaic lungs compared to the control lungs. **h** Volcano plots of DEGs between different cell clusters in *Lats1/2*-mosaic lungs at 17.5 *dpc*. DEGs in a given cluster carry the same color as shown in (**e**). The y-axis represents the average change in gene expression for a given gene between the two designated clusters. The x-axis represents the difference in the percentage of cells expressing the given gene between the two clusters in question. Changes in gene expression are statistically significant ($p < 0.05$) for all depicted DEGs (two-tailed Student's *t*-test). **i** A partial list of representative genes that were used to identify AT1 cells at four differentiation states. State 1: cells from *Lats1^f/f^; Lats2^f/f^; Sftpc^Cre/+^* lungs at 14.5 *dpc*. State 2: cells from control lungs at 17.5 *dpc*. State 3: cells from *Lats1^f/f^; Lats2^f/f^; Sftpc^Cre/+^* lungs at 17.5 *dpc*. State 4: cells from the control lungs of neonatal or adult mice. **j** Volcano plots of DEGs between two different AT1 states at the stages as indicated. DEGs in a given cluster carry the same color. The y-axis represents the average change in gene expression for a given gene between the two designated clusters. The x-axis represents the difference in the percentage of cells expressing the given gene between the two clusters in question. Changes in gene expression are statistically significant ($p < 0.05$) for all depicted DEGs (two-tailed Student's *t*-test).

*Elf5* (a *Fgf*-sensitive transcription factor), *Etv5*, *Sp5*, and *Tfcp2l1* (a grainyhead transcription factor), which have been reported to regulate lung development. For instance, loss of *Cebpa* is associated with dysfunction of AT2 cells[41]. *Etv5* is required for the maintenance of AT2 cells[42]. *Sp5* drives primary cilia formation[36]. Misexpression of *Elf5* was known to disrupt lung branching and differentiation[43]. Loss of *Tfcp2l1* enhanced AT2-AT1 differentiation in a mouse model of alveolar regeneration induced by influenza[44]. These examples illustrate that our scRNA-seq has identified a candidate cell population and their cellular trajectory for these regulators during alveolar fate determination. As described above, further analysis of other regulators and transcription factors identified in this study could reveal their new functions in lung development and cell fate determination.

To identify the YAP/TAZ targets and their regulators that control the AT1/AT2 fate in the alveolar epithelium, we performed multiomics analysis on isolated nuclei from control ($n = 7$) and *Lats1^f/f^; Lats2^f/f^; Sftpc^Cre/+^* (*Lats1/2*-mosaic) ($n = 7$) lungs at 17.5 *dpc*. snRNA-seq and snATAC-seq data from 2526 control and 3210 mutant lung nuclei were analyzed.

We identified differentially accessible regions (DARs) between AT1 and AT2 cells in control lungs at 17.5 *dpc* (Supplementary Fig. 19). Multiple motifs within DARs were revealed. The TEAD motifs were enriched in AT1 cells as expected. The other motifs enriched in either AT1 or AT2 cells provide insight into potential regulators of SOX9⁺ cell differentiation into AT1/AT2 cells in the alveolar epithelium. DEGs associated with AT1 cells contain multiple motifs (DARs) in their regulatory regions (Supplementary Fig. 19). Several DEGs associated with the transition between SOX9⁺ and AT1⁺ states contain the TEAD motif in their regulatory regions (Supplementary Fig. 19). This suggests that YAP/TAZ utilizes a regulatory network to control the transition of SOX9⁺ to AT1 cells.

### The developmental path of SOX9⁺ cells in mouse lungs can be mapped onto human lungs

Our molecular analysis of mouse lungs has produced the developmental paths and fate determination of SOX9⁺ cells (Figs. 6c, 7c). Taking advantage of the available scRNA-seq data on human fetal lungs[45], we asked if these paths could be mapped onto human lungs. We found that cluster 0 of control mouse lungs at 14.5 *dpc* expressed several markers that correspond to the early tip/early stalk of human lungs at 5 and 6.86 post-conception weeks (pcw), while cluster 2 can be identified with markers of the early stalk/early airway progenitor of human lungs (Supplementary Fig. 20a). Such correspondence was preserved in clusters 0 and 2 of *Lats1^f/f^, Lats2^f/f^, Sftpc^Cre/+^* (*Lats1/2*-mosaic) mouse lungs at 14.5 *dpc* (Supplementary Fig. 20b). For control and mutant mouse lungs at 17.5 *dpc*, cluster 0 corresponds to the late tip (and others) of human lungs at 22 pcw (Supplementary Fig. 20c, d). Interestingly, cluster 4 in control and mutant mouse lungs contains neuroendocrine cells, similar to those found in human lungs (Supplementary Fig. 20e, f). These results reveal parallel developmental paths of lung cells in mice and humans. It also highlights the importance of molecular analysis of mouse lungs to complement studies of human lungs.

## Discussion

Our study has advanced the mechanistic understanding of lung size control and cell fate specification (Fig. 8, Supplementary Fig. 22). Hippo signaling shapes lung development through its effects on SOX9⁺ progenitors. By controlling the expansion of a SOX9⁺ progenitor pool at the tip of lung branches, Hippo signaling regulates tip bifurcation and lung outgrowth, a key element in controlling lung size. Moreover, Hippo signaling is modulated in subpopulations of SOX9⁺ progenitors, regulating their differentiation through distinct sequences of transitional states toward cell fates in the conducting airways or the alveolar epithelium. These advances rely on the ability to generate a mosaic pattern of Hippo activity, coupled with transcriptome and chromatin accessibility analysis.

Our results suggest that Hippo signaling is tuned to a lower level in the distal SOX9⁺ subdomain. However, YAP/TAZ levels at the distal domain may display wave-like fluctuations in sync with repetitive branching morphogenesis. The molecular basis of Hippo signal modulation in different subdomains is unclear. This process likely involves additional signaling pathways and feedback controls. Analysis of chromatin accessibility and regulation of YAP/TAZ targets in subpopulations of SOX9⁺ progenitors could provide candidates for modulating the spatial activity of the Hippo pathway. Cells in different subdomains may regulate YAP/TAZ levels in response to local mechanical stress. In support of this notion, the composition and stiffness of the extracellular matrix vary along the proximo-distal axis of lung branches (Supplementary Fig. 21). These hypotheses require experimental validation.

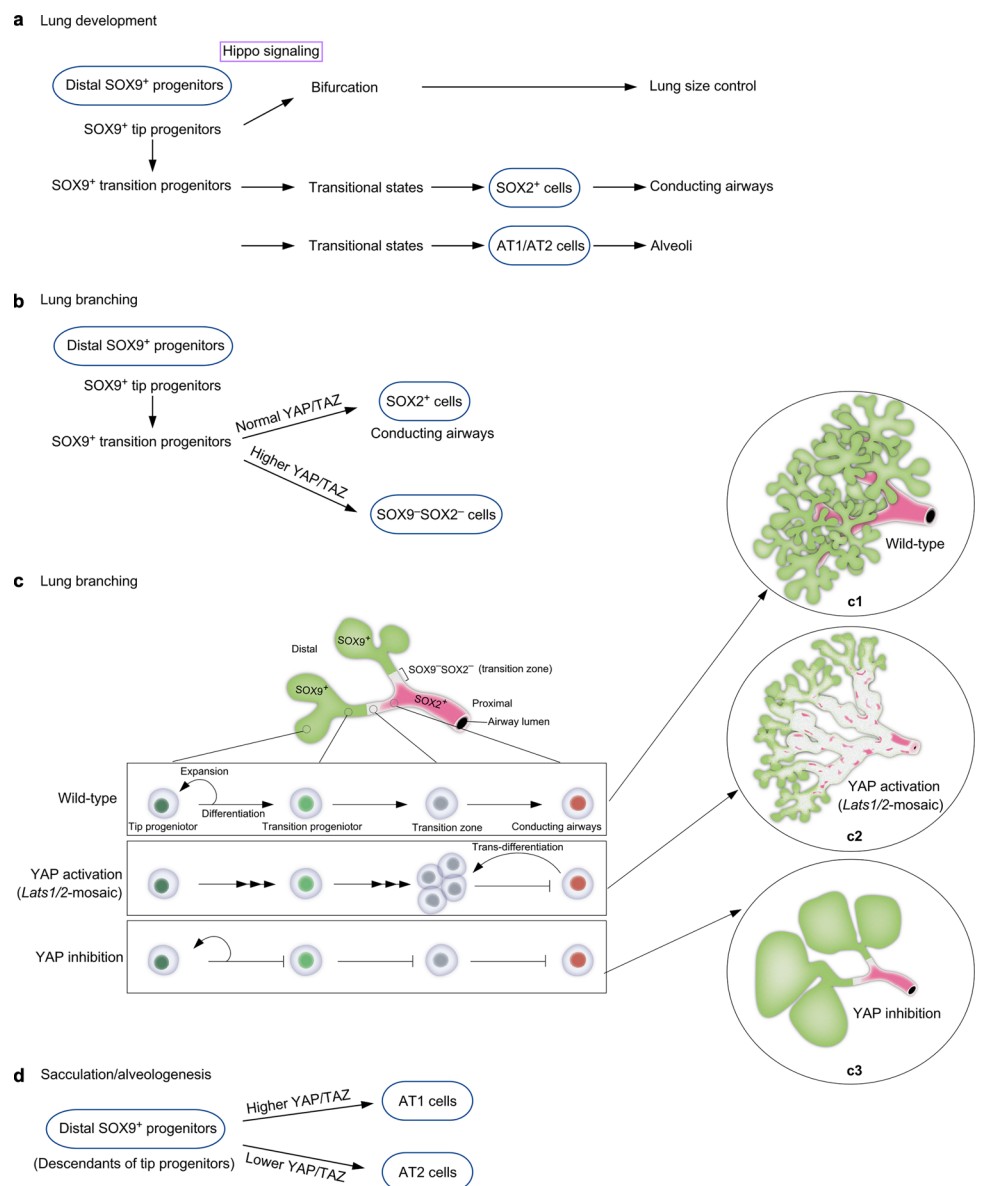

**Fig. 8 | A model of mouse lung development that is regulated by YAP/TAZ activity. a** Schematic diagram of the main findings of this study. Hippo signaling (YAP/TAZ levels) regulates the size and differentiation of the distal SOX9+ progenitors. Bifurcation at the distal tip underlies lung size control. YAP/TAZ levels also control the differentiation of SOX9+ progenitors, including tip progenitors and transition progenitors, through the transitional states to generate SOX2+ cells that form the conducting airways, and later to generate AT1/AT2 cells that produce alveoli. **b** Schematic diagram of the production of SOX2+ cells from the distal SOX9+ progenitors during lung branching through regulating YAP/TAZ levels. **c** Schematic diagram of a model in which YAP/TAZ levels at different positions along the proximo-distal axis of lung branches control cell proliferation and differentiation. In control lungs, the tip progenitors at the distal SOX9+ domain generate transition progenitors, which lie immediately proximal to the tip progenitors. The transition

progenitors produce cells in the transition zone, which subsequently generate proximal SOX2+ cells to form the conducting airways. Increased YAP/TAZ levels promote cell differentiation of tip progenitors toward transition progenitors and cells in the transition zone. SOX2+ cells in the presence of elevated YAP/TAZ levels transdifferentiate into cell fate in the transition zone. The diagram in (c2) depicts the changes in cells in *Lats1/2*-mosaic lungs described in this study, including the presence of residual tip progenitors, depletion of transition progenitors, and the mass production of cells with a transition-zone fate. Tip progenitors with reduced YAP/TAZ levels fail to differentiate into transition zone cells and SOX2+ cells. We speculate that the generation of transition progenitors is also blocked in this setting. **d** Schematic diagram of differentiating the distal SOX9+ progenitors toward the AT1 or AT2 fate by different YAP/TAZ levels during sacculation and alveologenesis.

Despite a significant reduction in the number of SOX9+ cells at the tip, dichotomous branching and lung size are largely preserved in *Lats1$^{f/f}$; Lats2$^{f/f}$; Sftpc$^{Cre/+}$* (*Lats1/2*-mosaic) lungs. It is possible that only a fraction of SOX9+ cells are utilized for dichotomous branching. Alternatively, a lower number of SOX9+ cells triggers a regulatory mechanism for size control of distal branches. This process may involve changes in cell proliferation/differentiation rates and the recruitment of additional signaling pathways. The molecular

machinery downstream of Hippo signaling to execute lung size control is unknown. Likewise, identifying YAP/TAZ targets and their regulators could be informative. The discovery that dichotomous branching is controlled through Hippo signaling in the distal SOX9+ progenitors sheds light on the mechanism of lung size control. With this insight, a combination of theoretical modeling, computer simulation, and 3D bioprinting can be useful for further testing the relationship between the SOX9+ progenitor pool size, the rates of proliferation

and differentiation, cell migration, and lung branch size in a 3D context.

In regions populated by SOX9⁻SOX2⁻ cells in *Lats1/2*-mosaic lungs, domain branching fails to occur. SOX9⁻SOX2⁻ cells display elevated YAP/TAZ activities. We surmise that the downstream events of YAP/TAZ activation, such as aberrant cell junctions and cell migration, disrupt the cell-cell interactions necessary for domain branching.

While the size of *Lats1/2*-mosaic lungs was mostly preserved, and dichotomous branching at the tips proceeded, saccules failed to be produced. We speculate that the number of SOX9⁺ cells at the tip, while adequate for bifurcation, is insufficient to generate saccules. Alternatively, a reduction in AT2 cells in *Lats1/2*-mosaic lungs contributes to defective sacculation. This is consistent with the observation that the appearance of AT2 cells is associated with the formation of the saccules and alveoli.

Our transcriptome and chromatin accessibility studies provide molecular insight into how subpopulations of SOX9⁺ progenitors undergo different sequences of transitional states to generate cell types in either the conducting airways or the respiratory epithelium. We have not only revealed how YAP/TAZ levels control this process but also identified candidate regulators of the transitional states. The function and transition of some of these cell states are supported by genetic analysis. Despite the new insight, our conclusions require experimental validations. For instance, these results form the basis of future investigations to reveal the spatial and temporal distribution of cell populations in a particular transitional state and understand how the transition of cell fates is regulated at the molecular level. Additional experiments are also required to probe the path, plasticity, and reversibility between cell states. In this regard, it is crucial to identify and test the functions of key transcription factors that determine different cell fates.

Our findings represent the first extensive report of transitional states from SOX9⁺ to SOX2⁺ cells to construct the conducting airway. Our earlier study supports a model in which SOX9⁺ cells can be functionally categorized as tip progenitors and transition progenitors[7,46]. We anticipate that further transcriptomic analysis will define the molecular signatures of tip and transition progenitors and their descendants during branching and sacculation. The trajectories of two possible paths to reach the SOX2⁺ state may reflect the plasticity of the SOX9⁺ to SOX2⁺ transition in a given SOX9⁺ cell. Another possibility is that SOX9⁺ cells at different locations along the proximo-distal axis of lung branches follow different paths to become SOX2⁺ cells.

Our results also depict the first extensive characterization of the transitional states from SOX9⁺ to AT2 or AT1 cells to generate alveoli. In this setting, only one trajectory was revealed. Analysis of cells expressing AT1 markers in control and *Lats1/2*-mosaic lungs at 14.5 and 17.5 *dpc* reveals a path of AT1 maturation associated with transcriptome changes. We envision that further single-cell analysis will uncover additional intermediate steps and their regulators for AT1 maturation during embryonic and postnatal lung development. AT1 cells are produced from AT2 cells following lung injury. During the repair process, the transitional states that connect AT2 to AT1 cells have been reported[47,48]. A fundamental question is whether the production of AT1 cells charts a common path during development and following lung injury. Our analysis delineates several maturation states of AT1 cells during development, enabling comparative studies on AT1 cell production following lung injury. Even if the production of AT1 cells adopts different paths during development and following lung injury, both can be exploited to stimulate AT1 cell generation in adult lungs.

Loss of *Lats1/2* in the progeny of SOX9⁺ cells results in a failure to produce SOX2⁺ cells to form the conducting airways. In fact, these cells also lose *Sox9* expression. Inactivation of *Lats1/2* in SOX2⁺ cells results in the termination of *Sox2* expression. Despite multiple differentiation trajectories to reach the SOX9⁻SOX2⁻ state, SOX9⁻SOX2⁻ cells turn on

the fate of AT1 cells once Hippo signaling is active. The molecular mechanisms underlying changes in cellular properties are unclear. Addressing this issue also requires identifying and testing the function of key regulators downstream of YAP/TAZ that control cell fates.

SOX9⁺ cells are present in human lungs[45]. Organoids derived from human fetal lungs display progenitor properties. Our analysis of mouse and human scRNA-seq data suggests that at least one common path can be identified in both species. The path of cluster 0 to 2 in mice is associated with the production of AT1 cells in *Lats1/2*-mosaic lungs. This raised the question of whether the SOX9⁺ cluster in the mouse lungs at 14.5 *dpc* can be subgrouped based on their transcriptomes, with different SOX9⁺ subsets as the precursors for the conducting airways and alveolar epithelium[49–51]. Differences likely exist in the stepwise differentiation program between mouse and human lungs. In both systems, the signaling pathways and transcription factors used to specify cell types are not fully understood. The eventual validation of fate determination models inferred from human lung organoids would need to await genetic and molecular studies in non-human primates. In this regard, our mouse work provides key experimental settings that complement functional studies on human lungs.

Together, our work has provided a new conceptual framework for understanding how Hippo signaling controls the distinct steps of SOX9⁺ cell proliferation and differentiation. This information lays the groundwork for further functional studies. It also provides a platform to investigate how other signaling pathways interact with Hippo signaling during lung growth and fate specification in mammals.

## Methods

### Animal husbandry

All mouse experiments in this study were performed in accordance with protocols approved by the Institutional Animal Care and Use Committee (IACUC) of the University of California, San Francisco (UCSF). Rodents were housed in a standard specific pathogen-free (SPF) facility at standard ambient temperature and humidity, with a 12:12-hour light-dark cycle. Mouse matings were set up to obtain animals with the genotypes described in this study. Information on mouse ages, numbers, and genotypes was included in the text and figures. N pairs in the figure legends denote n controls and n mutants, which were collected from as many litters as needed to reach that number. All the controls used in this study are littermate controls. Their genotypes vary depending on the crosses used in a given experiment. Mice were maintained on a mixed genetic background. Sex was not considered in the study design since sex is not a factor in influencing the outcome of lung development in our studies.

The mouse strains used in this study and their sources are listed below. *Yap* floxed allele (*Yap^f*) [*Yap1^tm1.1Eno*; MGI: 5446483; Dr. Eric Olson], *Taz* floxed allele (*Taz^f*) [*Wwtr1^tm1.1Eno*; MGI: 5544289; Dr. Eric Olson], *Lats1* floxed allele (*Lats1^f*) [*Lats1^tm1.1Jfm*/RjoJ; MGI: 5568586; RRID: IMSR_JAX:024941], *Lats2* floxed allele (*Lats2^f*) [*Lats2^tm1.1Jfm*/RjoJ; MGI: 5568589; RRID: IMSR_JAX:025428], *ROSA26^mTmG* [B6.129(Cg)-*Gt(ROSA)26Sor^tm4(ACTB-tdTomato,-EGFP)Luo*)/J; MGI: 3803814; RRID: IMSR_JAX:007676], *Shh^Cre* [B6.Cg-*Shh^tm1(EGFP/cre)Cjt*)/J; MGI: 3619470; RRID: IMSR_JAX:005622], *Spc^Cre* (*Sftpc^Cre*) [*Tg(Sftpc-cre)1Blh*; MGI:3574949; Dr. Brigid Hogan], *Sox9^Cre* [*Sox9^tm3(Cre)Crm*; MGI: 3608931; Dr. Benoit de Crombrugghe], *Sox9^CreER* [C57BL/6-*Sox9^em1(cre/ERT2)Tchn*; MGI: 6386787; RRID: IMSR_JAX:035092], *Sftpc^CreER* [*Sftpc^tm1.1(cre/ERT2)Ptch*; MGI: 5485991; Dr. Pao-Tien Chuang].

### Tamoxifen administration

The stock solution of tamoxifen (50 mg/ml) was prepared by dissolving tamoxifen in corn oil. To prepare a working solution of tamoxifen (10 mg/ml), the stock solution was diluted 1:5 in corn oil. In general, 75-100 µl of the working solution was delivered to pregnant female mice through intraperitoneal injection. However, different mouse CreER lines may require varying amounts of tamoxifen to achieve efficient

recombination. Detailed information on the stages of tamoxifen injection and lung collection was provided in the figures or figure legends.

## Whole-mount immunofluorescence

Embryonic mouse lungs were harvested at the indicated time points and fixed in 4% paraformaldehyde (PFA) in PBS on ice for 1 h. The lungs were washed with 0.1% Tween-20/PBS for 30 min. Samples were then treated with graded methanol solutions (25%, 50%, 75%, 100%) and then incubated in 5% $H_2O_2$/methanol overnight. The next day, the samples were rehydrated through a series of graded methanol solutions (100%, 75%, 50%, 25%, and 0%), which were diluted in 0.1% Tween-20/PBS. After incubation in blocking buffer (1.5% BSA/0.5% Triton X-100/PBS) for 6 h, the samples were then incubated with primary antibodies at 4 °C overnight. The next day, samples were washed with blocking buffer for 5 h and then incubated with secondary antibodies at 4 °C overnight. A Nikon Eclipse E1000 microscope with a SPOT 2.3 CCD camera was used to capture the images.

The primary antibodies for whole-mount immunostaining were rat anti-E Cadherin (1:200, Life Technologies, Cat# 13-1900; RRID:AB_2533005), rabbit anti-phospho-YAP (Ser127) (1:100, Cell Signaling, Cat# 4911S; RRID:AB_2218913), rabbit anti-SOX2 (D9B8N) (1:200, Cell Signaling Technology, Cat# 23064S; RRID:AB_2714146), goat anti-SOX9 (1:200, R&D Systems, Cat# AF3075; RRID:AB_2194160).

## Immunofluorescence and Immunohistochemistry

Dissected mouse lungs were fixed in 4% PFA in PBS for 1-2 h at 4 °C. Lungs were embedded in paraffin wax and sectioned at 6 μm or embedded in OCT and sectioned at 6 μm. Histological analysis was performed using standard hematoxylin and eosin staining procedures[46,52,53].

For immunostaining[46,52,53], sections were dewaxed with xylene (VWR 89370-088) then rehydrated with graded alcohol solutions (100% ethanol, 95% ethanol, 80% ethanol; Decon V1001). Antigen retrieval was performed by microwaving sections in 0.1 M Sodium Citrate, pH 6.0 (Fisher BP327) for 6.5 min, then cooled to room temperature (RT). Sections were permeabilized with 0.2% Triton X-100 (MilliporeSigma T9284) in PBS at RT for 10 min, then blocked with 3% bovine serum albumin (BSA; MilliporeSigma A2153)/0.02% Triton X-100/PBS at RT for 1 h prior to incubation with primary antibodies diluted in blocking solution at 4 °C overnight. Sections were rinsed with 0.02% Triton-X100/PBS, then incubated with secondary antibodies diluted in blocking solution at RT for 45 min, followed by amplification, if necessary. Nuclei were stained with DAPI (1:10,000 dilution; MilliporeSigma D9542) and mounted with VECTASHIELD Antifade Mounting Medium (Vector Laboratories H1000).

The primary antibodies used include: rabbit anti-NKX2.1 (1:100, Epitomics, Cat# 2044-1; RRID:AB_1310784), chicken anti-GFP (1:200, Aves Labs, Cat# GFP-1010; RRID:AB_2307313), rabbit anti-prosurfactant protein C (proSP-C) (1:200, Seven Hills Bioreagents, Cat# WRAB-9337; RRID:AB_2335890), Syrian hamster anti-T1a (1:200, Developmental Studies Hybridoma Bank, Cat# 8.1.1; RRID:AB_531893), mouse anti-HOPX (E-1) (1:100, Santa Cruz Biotechnology, Cat# sc-398703; RRID:AB_2687966), goat anti-CC10 (1:200, Santa Cruz Biotechnology, Cat# sc-9773; RRID:AB_2183391), rabbit anti-CCSP (1:200, Seven Hills Bioreagents, Cat# WRAB-3950; RRID:AB_451716), mouse anti-acetylated tubulin (1:200, Santa Cruz Biotechnology, Cat# sc-23950; RRID:AB_628409), mouse anti-α-actin (ACTA2) (SMA), clone 1A4 (1:200, Santa Cruz Biotechnology, Cat# sc-32251; RRID:AB_262054), rat anti-PECAM-1 (CD31) (1:150, Santa Cruz Biotechnology, Cat# sc-18916; RRID:AB_627028), rabbit anti-PDGFRA (1:150, Cell Signaling Technology, Cat# 3164S; RRID:AB_2162351), goat anti-PDGFRB (1:200, R&D Systems, Cat# AF1042; RRID:AB_2162633), rabbit anti-LATS1 (C66B5) (1:100, Cell Signaling Technology, Cat# 3477S; RRID:AB_2133513), rabbit anti-Phospho-YAP (Ser127) (1:100, Cell

Signaling Technology, Cat# 4911S; RRID:AB_2218913), mouse anti-YAP (63.7) (1:100, Santa Cruz Biotechnology, Cat# sc-101199; RRID:AB_1131430), rabbit anti-YAP1 (1:100, Novus Biologicals, Cat# NB110-58358; RRID:AB_1849483), goat anti-CTGF (1:100; Santa Cruz Biotechnology #sc-14939; RRID:AB_638805).

Secondary antibodies and conjugates used were donkey anti-rabbit Alexa Fluor 488 (1:1000, Life Technologies, Cat# A-21206; RRID:AB_2535792) or 594 (1:1000, Life Technologies, Cat# A-21207; RRID:AB_141637), donkey anti-chicken Alexa Fluor 488 (1:1000, Jackson ImmunoResearch, Cat# 703-546-155; RRID:AB_2340376) or 647 (1:1000, Jackson ImmunoResearch, Cat# 703-606-155; RRID:AB_2340380), donkey anti-mouse Alexa Fluor 488 (1:1000, Life Technologies, Cat# A-21202; RRID:AB_141607) or 594 (1:1000, Life Technologies, Cat# A-21207; RRID:AB_141637), and donkey anti-rat Alexa Fluor 594 (1:1000, Life Technologies, Cat# A-21209; RRID:AB_2535795). The biotinylated secondary antibodies used were goat anti-hamster (1:1000, Vector Laboratories, Cat# BA-9100; RRID:AB_2336137), donkey anti-rabbit (1:1000, Jackson ImmunoResearch, Cat# 703-066-155; RRID:AB_2340355), donkey anti-rat (1:1000, Jackson ImmunoResearch, Cat# 712-065-150; RRID:AB_2340646), and horse anti-mouse (1:1000, Vector Laboratories, Cat# BA-2000; RRID:AB_2313581). The signal was detected using streptavidin-conjugated Alexa Fluor 488 (1:1000, Jackson ImmunoResearch, Cat# 016-540-084; RRID:AB_2337249), 594 (1:1000, Jackson ImmunoResearch, Cat# 016-580-084; RRID:AB_2337250), or 647 (1:1000, Jackson ImmunoResearch, Cat# 016-600-084; RRID:AB_2341101) or HRP-conjugated streptavidin (1:1000, Jackson ImmunoResearch, Cat# 016-030-084; RRID:AB_2337238) coupled with fluorogenic substrate Alexa Fluor 594 tyramide for 30 s (1:200, TSA kit; Akoya Biosciences, Cat# NEL753001KT) or Cyanine 555 (1:200, Biotium, Cat# 96020) for 30 s.

Confocal images were captured on a Leica SPE laser-scanning confocal microscope. Adjustments to the red/green/blue/grey histograms and channel merges were performed using LAS AF Lite (Leica Microsystems).

## Cell proliferation assays

Cell proliferation in embryonic lungs was assessed by EdU incorporation. Briefly, 1 h prior to lung collection, 300 μl of EdU (Biosynth, Cat# NE08701)/PBS solution (5 mg/ml) was injected intraperitoneally into pregnant female mice at the indicated time points. EdU incorporation was evaluated by the Click-iT EdU Alexa Fluor 488 Imaging Kit (Thermo Fisher, Cat# C10337). To measure the proliferation rate of SOX9+, SOX2+, SOX9−SOX2− cells, EdU spots were co-stained with SOX2 or SOX9 on adjacent sections, where DAPI was used to identify individual lung cells. The signals of EdU, SOX2, and SOX9 all reside in the nucleus and also co-localize with the DAPI signal. The number of proliferating SOX2+ cells (EdU+SOX2+DAPI+), proliferating SOX9+ cells (EdU+SOX9+DAPI+), proliferating SOX9−SOX2− cells (EdU+SOX9−SOX2−DAPI+), the total number of SOX2+ cells (SOX2+DAPI+), the total number of SOX9+ cells (SOX9+DAPI+), and the total number of SOX9−SOX2− cells (SOX9−SOX2−DAPI+) were manually counted using ImageJ. The proliferation rates of SOX2+ cells, SOX9+ cells, and SOX9−SOX2− cells were calculated as the ratios of (EdU+SOX2+)/(SOX2+), (EdU+SOX9+)/(SOX9+), and (EdU+SOX9−SOX2−)/(SOX9−SOX2−). For each biological replicate, at least 5 sections were used for quantification.

## Quantification assays

Quantification of immunostaining signals was performed on images of control and $Lats1^{f/f}$; $Lats2^{f/f}$; $Sftpc^{Cre/+}$; $ROSA26^{mTmG/+}$ ($Lats1/2$-mosaic) lungs at 14.5 $dpc$ under the same exposure. A square box of the same size was placed on the distal and proximal lung epithelium stained with anti-pYAP. ImageJ was used to measure and calculate the relative signal intensity of pYAP staining.

Quantification of the relative SOX9⁺ area in each distal domain or distal bud was performed by ImageJ. Likewise, the quantification of the SOX2⁺ domain within the domain containing both SOX2⁺ and SOX9⁻SOX2⁻ cells was performed by measuring the areas for each domain using ImageJ. The relative length of the SOX9⁻SOX2⁻ domain refers to the relative length between control and *Lats1/2*-mosaic lungs measured in ImageJ. The SOX9⁻SOX2⁻ domain is bounded proximally by the SOX2⁺ domain and distally by the SOX9⁺ domain.

The branching pattern was traced on *Sftpc*^Cre/+^; *ROSA26*^mTmG/+^ (control) and *Lats1*^f/f^; *Lats2*^f/f^; *Sftpc*^Cre/+^; *ROSA26*^mTmG/+^ (*Lats1/2*-mosaic) lungs following the published method[1]. All the lineages, including domain branching and bifurcation from branch L.L1 until 14.5 *dpc*, were quantified. Epithelial counting relied on both the *ROSA26*^mTmG^ reporter line and SOX2/SOX9 whole-mount immunostaining.

## Bulk RNA-Seq analysis

Bulk RNA-seq was performed as follows[46]. Briefly, the embryonic lungs were dissected from control and *Lats1*^f/f^; *Lats2*^f/f^; *Sftpc*^Cre/+^ mice at 13.5 *dpc* and lysed in 0.5 ml TRIzol (Life Technologies). 100 μl chloroform was then added to the lysates. The samples were then centrifuged at 4 °C for 15 min. The upper aqueous layer was collected and mixed with an equal volume of 70% ethanol. RNAs were extracted with the RNeasy Mini Kit (Qiagen) following the manufacturer's instructions. RNA quality was evaluated using an Agilent 2100 Bioanalyzer. Samples were sequenced on an Illumina HiSeq. Differential gene expression, gene ontology (GO) enrichment analyses, and the barplot of gene ontology enrichments were performed using RStudio. Heatmap images were generated using the online Heatmapper software.

## qPCR analysis

qPCR analysis was performed using a QuantStudio 5 PCR System (Applied Biosystems). The following primers for mouse genes were used: *Gapdh* (forward, 5′-AGGTTGTCTCCTGCGACTTCA-3′; reverse, 5′-CCAG GAAATGAGCTTGACAAAGTT-3′), *Ctgf* (forward, 5′-CTCCACCC GAGTTACCAATG-3′; reverse, 5′-TGGCGATTTTAGGTGTCCG-3′), *Ajuba* (forward, 5′-GGTATCTATGGGCAGAGCAATG-3′; reverse, 5′- CAGTAG ACAGAGCCATTGACG-3′), *Cyr61* (forward, 5′-GTGAAGTGCGTCCTTGT GGACA-3′; reverse, 5′-CTTGACACTGGAGCATCCTGCA-3′).

## Lung cell dissociation

Lungs were harvested and dissociated as follows[54]. Briefly, mouse lungs were dissected from embryos and rinsed in DPBS (Gibco 14190-144). Tissues were minced into small pieces using scissors or razor blades and then transferred to 1.5 ml tubes containing 0.5 mL Dissociation Buffer (0.05% Collagenase B (Roche 11088835001), 1.2 U/mL Dispase II (Gibco 17105-041), 80 Kunitz units/mL DNase I (MilliporeSigma DN25)). The samples were incubated at 37 °C for 30–50 min and triturated every 10 min using a P-1000 pipette. Cells were washed with 1 mL PBS and then resuspended in 90% fetal bovine serum (FBS; Phoenix Scientific PS-100) + 10% DMSO (MilliporeSigma D2650). Cells were stored at −80 °C overnight and then transferred to liquid nitrogen for long-term storage.

Cells were thawed in a 37 °C water bath for 2 min and then washed with 1 mL 10% FBS/Leibovitz L-15 media (Gibco 21083-027). The cells were resuspended in 100 μL DNase I (1600 Kunitz units/mL) and then washed with 1 mL DPBS. The cells were passed through a 70 μm strainer (Bel-Art Products H13680-0070) and resuspended in 200 μL 0.5% bovine serum albumin (BSA; Miltenyi Biotec 130-091-376).

## Sample preparation for scRNA-seq and multiomics analysis

For scRNA-seq and multiome (snRNA-seq/snATAC-seq) experiments, mouse lung cells were incubated with eFluor 450 anti-CD45 (eBioscience 48-0451-80) and eFluor 450 rat anti-TER-119 (eBioscience 48-5921-80) for 30 min at 4 °C. Cells were washed with 1 mL 0.5% BSA, then passed through a 40 μm strainer (Bel-Art Products

H13680-0040) and resuspended in 0.5% BSA + 0.5 μg/mL DAPI (MilliporeSigma D9542) before sorting on a BD FACSAria Fusion flow cytometer.

Multiome samples were prepared following the 10x Genomics protocol. Briefly, 50,000-100,000 sorted cells were incubated in 0.1X Lysis Buffer (10 mM Tris-HCl (pH 7.4), 10 mM NaCl, 3 mM MgCl₂, 0.1% Tween-20, 0.1% Nonidet P40 Substitute, 0.001% Digitonin, 1% BSA, 1 mM DTT, 1 U/μL RNase inhibitor in nuclease-free water) on ice for 3 min. The samples were washed three times with Wash Buffer (10 mM Tris-HCl (pH 7.4), 10 mM NaCl, 3 mM MgCl₂, 0.1% Tween-20, 1 mM DTT, 1 U/μL RNase inhibitor in nuclease-free water), then resuspended in Diluted Nuclei Buffer prior to library preparation.

## scRNA-seq analysis

scRNA-seq was performed using the GEM-X Universal 3′ Gene Expression kit (10x Genomics) following the manufacturer's instructions.

Sequencing reads from each single-cell RNA-seq (scRNA-seq) sample were preprocessed, aligned, and quantified using the CellRanger pipeline (version 8.0.1, 10x Genomics) with the mm10 mouse reference genome (refdata-gex-mm10-2020-A, 10x Genomics).

Seurat R package (version 5.2.1) was used for quality control (QC), normalization, scaling, and identification of variable features for each sample. Cells with 200–500 Unique Molecular Identifiers (UMIs) mapped to unique genes, and with less than 5% of UMIs mapped to mitochondrial genes, were retained. EmptyDrops was used to generate a barcode rank plot of UMI count and a fitted spline, which was used for detecting the knee point. Barcodes detected as putative cell-containing droplets at a false discovery rate (FDR) of 0.1% or less were kept. Clusters that expressed lineage markers for hematopoietic, endothelial, mesenchymal, or mesothelial cells were removed. Highly variable genes for dimensionality reduction were selected using the Seurat::FindVariableGenes function (nfeatures = 2000). Principal components (PCs) were identified using the Seurat::RunPCA function. The number of PCs accounting for more than 90% of the cumulative variation in each sample was identified for downstream analysis. A K-nearest neighbor (KNN) graph was built using the Seurat::FindNeighbors function. Graph-based clustering of the PC analysis was done using the Seurat::FindClusters function (Louvain algorithm). Projection of the cells on a 2D map was performed with uniform manifold approximation and projection (UMAP) using the Seurat::RunUMAP function. UMAP plots overlaid with gene expression levels were generated using the Seurat::FeaturePlot function. Dot charts for gene expression were generated using the Seurat::DotPlot function.

Differentially expressed genes (DEGs) were identified using the Seurat::FindMarkers function (test.use = "wilcox"). Markers with a false discovery rate (FDR) less than 0.05 were kept. Volcano plots for DEGs between clusters/samples were generated using ggplot2 (v3.5.1) and ggrepel (v0.9.6). The x-axis represents the difference in the percentage of cells expressing the gene of interest between the two clusters/samples; the y-axis shows the average fold change in gene expression between the two clusters/samples.

Pseudotemporal analysis was done using the Monocle3 R package (v1.3.7). Monocle3 was used to learn the sequence of gene expression changes using the monocle3::learn_graph function. Cells were then ordered according to their progression through the developmental program using the monocle3::order_cells function. Predicted pseudotime values for each cell were overlaid onto the previously generated UMAP projection.

The SCENIC (v0.12.1) pipeline was used to infer co-expression modules from single-cell expression matrices of EPCAM⁺ cell clusters. The cisTarget database (mc_v10_clust) and Motif2TF annotations database (v10) (Aerts lab) were used for motif enrichment in gene sets. Transcription factor (TF)–target gene adjacency lists were pruned to retain transcription factors with highly enriched motifs (within the top 300 ranked motifs) around the predicted transcription start site (TSS)

of their target genes. The search space is within 500 base pairs upstream and 100 base pairs downstream of the TSS of the target gene. Gene regulatory networks (GRNs) were constructed using previously predicted TF–target connections within the differentially expressed genes (DEGs) between two cell clusters, based on pruned TF–target gene lists. Pearson correlation coefficients were used to assess the direction of the relationship between TFs and their target genes. GRN plots were generated using the RStudio package igraph (v2.1.4) and ggplot2 (v3.5.1).

Merging and integration of scRNA-seq data of multiple samples were performed as follows. For publicly available data, EPCAM+ cells were aggregated, and clusters that expressed lineage markers for hematopoietic, endothelial, mesenchymal, and mesothelial cells were removed. Highly variable genes for dimensionality reduction were selected using the Seurat::FindVariableGenes function (nfeatures = 2000). The batch effect was corrected using the Fast Mutual Nearest Neighbors Integration (Fast MNN Integration) method in the Batchelor (v1.24.0) package. A K-nearest neighbor (KNN) graph was built using the Seurat::FindNeighbors function. Graph-based clustering of the Fast MNN-reduced data was performed using the Seurat::FindClusters function (Louvain algorithm). Projection of the cells on a 2D map was performed with uniform manifold approximation and projection (UMAP) using the Seurat::RunUMAP function. Cell annotation was performed using LungMAP mouse lung Cellref as the reference (LungMAP_MouseLung_CellRef.v1.1). Anchors were identified using Seurat::FindTransferAnchors. Labels were transferred from reference and added to query object metadata using Seurat::TransferData and Seurat::AddMetaData, respectively. Similarly, label transfer was performed from the control to the *Lats1/2*-mosaic mutant data. Cells with a predicted score (predicted.score.max) of more than 0.8 in *Lats1/2*-mosaic mutants were identified as highly similar to control cells.

## Multiomics analysis

Multiomics was performed using the Chromium Next GEM Single Cell Multiome ATAC + Gene Expression (10x Genomics) following the manufacturer's instructions.

Raw sequencing reads were processed using the 10x Genomics Cell Ranger Arc (v2.0.2) pipeline. Reads were aligned with cellranger-arc count to the mm10 mouse reference genome. For downstream analysis, both the Signac R package (v1.14.0) and the Seurat R package (v5.2.1) were used. For snATAC-seq, nuclei with a read count more than 100,000, or less than 1800, or nucleosome signals more than 2, or with transcription factor enrichment scores less than 1 were excluded. For snRNA-seq, nuclei with a read count of more than 25,000 or less than 1000 were excluded. For epithelial cell analysis, we aggregated EPCAM+ nuclei, and clusters that expressed lineage markers for hematopoietic, endothelial, mesenchymal, and mesothelial nuclei were removed. Variable genes for dimensionality reduction were selected using Seurat::FindVariableGenes (nfeatures = 2000) for the snRNA-seq assay. Variable peaks for dimensionality reduction were selected using Signac::FindTopFeatures of the snATAC-seq assay. Dimensionality reduction of snATAC-seq was done with the Latent Semantic Indexing (LSI) method. Multimodal data from snATAC-seq and snRNA-seq were integrated using the Signac::FindMultiMoDalNeighbors function to construct a joint nearest-neighbor (JNN) graph using information from both assay modalities for clustering and visualization. Visualization of the clusters on a 2D map was performed with uniform manifold approximation and projection (UMAP) using the Seurat::RunUMAP function. Differentially expressed genes of each cluster were identified using the Seurat::FindAllMarkers function. Differentially accessible peaks for the desired group of nuclei were identified using the Seurat::FindMarkers function with the Likelihood Ratio Test (LR test), requiring a minimum presence in the tested group of nuclei of at least 5%. The peaks were considered significantly differential if they had a p-value of less than 0.005 and were present in at least 20% of the tested group of nuclei. Visualization of the peaks was done using the Signac::CoveragePlot function.

Enriched motifs within the test group of cells were identified using the Signac::FindMotifs function. Specific peaks within the differentially accessible regions (DARs) where each motif is enriched were identified, separated, and annotated to the nearest transcription start site (TSS) using the annotatePeak function from the ChIPseeker package. Peak annotations were then matched to the top 10% (fold change) of differentially expressed genes (DEGs) or transcription factors (TFs) within the DEGs between the test groups.

## Statistics and reproducibility

All biological repeats were three or more, and the number of biological replicates (n) was indicated in the figures and legends. For immunostaining data, at least three independent repeats were performed, yielding similar results. The micrographs shown in the figures are representative images. All statistical comparisons between groups were presented as mean ± SEM. Two-tailed Student's *t*-tests, one-way ANOVA, or two-way ANOVA were applied to calculate the P values, and the statistical significance was evaluated as * $P < 0.05$, ** $P < 0.01$, and *** $P < 0.001$. The lung phenotypes analyzed in this study are completely penetrant.

## Reporting summary

Further information on research design is available in the Nature Portfolio Reporting Summary linked to this article.

## Data availability

All the related data for this study are available in the published article and the supplementary information file. Raw and analyzed RNA-Seq data have been deposited in the NCBI Gene Expression Omnibus (GEO) database and are accessible via the GEO Series (GSE) accession numbers GSE269537, GSE319370, and GSE324638. Source data are provided with this paper.

## Code availability

The code used in the scRNA-seq and multiomics analysis is available on GitHub with the link https://github.com/yzaher/HippoSignaling.

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

## Acknowledgements

We thank Tatsuya Tsukui for invaluable suggestions on experimental procedures; Jianying Li and Wanpeng Wang for technical assistance; Kaveh Ashrafi, Andy Chang, Peng He, and Tatsuya Tsukui for their insightful comments on the manuscript. Some data for this study were acquired at the Nikon Imaging Center at CVRI. This work was supported by grants (R01 HL157446) from the National Institutes of Health to P.T.C.

## Author contributions

K.Z., M.B., Y.Z., E.Y., and P.T.C. designed the study. K.Z., M.B., Y.Z., E.Y., S.A.W., and T.A. performed the experiments. K.Z., M.B., Y.Z., E.Y., S.A.W., and T.A. analyzed the data. K.Z., Y.Z., and P.T.C. wrote the manuscript. P.T.C. supervised the research.

## Competing interests

The authors declare that they have no competing interests.
