## [Peer Review file · Nature Communications]

Hippo signaling differentially regulates distal progenitor subpopulations and their transitional states to construct the mammalian lungs

Corresponding Author: Dr Pao-Tien Chuang

Version 0:

Reviewer comments:

Reviewer #1

(Remarks to the Author)

Summary

Zhang et al. investigate the role of Hippo signaling in regulating the expansion and differentiation of the distal epithelial (Sox9+) progenitor domain and neighboring Sox9-/Sox2- in lung development. Although the Hippo pathway has been previously implicated in lung morphogenesis and epithelial fate specification, its precise role in controlling distal epithelial fate transitions remains unclear. In this study, the authors employ mosaic knockout mouse models to selectively disrupt Lats1/2 and Yap/Taz, enabling a heterogeneous perturbation of Hippo signaling. They concluded that hippo signaling regulates epithelial proximal to distal patterning, and further that these perturbations disrupt epithelial cell fate differentiation. The mouse genetic experiments are a tour de force, and the observed differentiation defects are compelling, making this work worthy serious consideration at Nature Communication. However, the manuscript needs some major improvement over the clarity and rigor.

Major Concerns

1. Given the mosaic nature of the Cre-mediated recombination, it is unclear whether the epithelial differentiation and lung morphogenesis defects result from the cell-autonomous effect of the knockout or cell non-autonomous effects, making it difficult to interpret the phenotypes. The authors should provide single-cell level characterization of the mosaic lung driven by the SFTPC promoter, to correlate the genotype with phenotype.
 - a. It is unclear if mutant cells within the mosaic lung have expected Hippo pathway changes. The authors should show the colocalization of Cre recombination (mTmG GFP channel) and pYAP to make this point.
 - b. It is unclear how mutant cells were distributed among Sox9+ and Sox2+ domains. This made it difficult to interpret if the outgrowth of the Lats1/2-mosaic lungs was due to compensation by the remaining wildtype cells, cell sorting, or a remaining function of the mutant cells. While these models are difficult to tease apart, a clear characterization of mutant cell position within Sox2/Sox9 domains would help narrowing the likely models. The authors can costain GFP (from the successful recombined mTmG), Sox9, and Sox2. Imaging data should be provided in the figure, rather than just a summary cartoon.
 - c. The analysis on the scRNA-sequencing on mosaic tissue does not separate out mutant and wildtype cells. If feasible, it would be more informative to identify the mutant cells in the data using a combination of Hippo pathway genes (taken together to avoid dropout effects) and provide direct comparison between mutant and wildtype cells.
2. Overall, I found the morphological defects in the Lats1/2-mosaic lungs are unsupported or overstated. For example, "multilayer epithelium" is a striking and unusual phenotype, but no imaging data was provided to support this claim. The absence of domain branches in the field of view is difficult to interpret, as it could be due to larger patterning issues that are not specific to the domain branching program. The large-scale airway architecture is constructed by domain branches, which appears generally normal, suggesting the classically defined "domain branching" process is maintained in the mutants. Authors should either avoid using "domain branching" when discussing the morphological defects, or provide additional morphological analysis to support the defect being specific to a branching mode.
3. The authors do not untangle the early effects of Lats1/2 on Sox9/Sox2 patterning, and how disruption to this patterning can compound later cell differentiation phenotypes. Specifically, in Fig. 5, the timepoints chosen are too early in development for the motivation authors presented (to separate earlier patterning defects from later cell differentiation phenotypes). The experiment needs to be redone with tamoxifen being dosed just before cell differentiation, which takes place at E17.5. Even with accounting for potential time from dose to recombination, tamoxifen should be applied no earlier

than E16. Ideally, authors should also provide a control that shows no major Sox9/Sox2 patterning disruptions just before E17.5 in samples with tamoxifen dosed at their chosen application timepoint. Additionally, it is unclear within the differentiated cells shown to be skewed towards AT1 fate, how many of these are mutant (again because of the mosaic nature of the mouse). Authors could address this by providing the GFP channel (in mTmG mice).

4. I found it difficult to match the annotation of the clusters and trajectories in the scRNA-sequencing data analysis with what is previously reported in the field (PMID: 39667932, 33707239). We suggest that authors annotate populations with matched names to previous studies, or provide clear parallels in the text between clusters and previously reported populations.

5. All conclusions made based on lineage and gene regulatory analysis are hypothetical without experimental validation. The authors should be careful with the language used to avoid overstatements.

6. Human data analysis as presented does not add knowledge to Hippo signaling in Sox9+ progenitors. We suggest that it is removed to streamline the message of the paper.

7. Currently, the paper structure makes it difficult to identify the key points and often restates ideas out of order. We suggest the following logical sequence to ground the reader in the main message and highlight the impressive results that may otherwise be overlooked: First, characterize the Lats1/2 mutant with a focus on cell differentiation (the later stages after E17.5), with the striking rescue cell differentiation result as well; Next, work backwards in developmental time and show how Sox9/Sox2 and transitional progenitor domains are affected; Then ask if cell fate is affected when the Sox9 domain is not messed up prior (proposed fix to fig 5); Finally, combine and streamline points in Fig 4+6.

Minor Comments

Fig. 1d-1i – It is hard to tell if pYAP levels are actually reduced. Can the authors provide clearer images and potentially quantification

Pg. 6 bottom (and pg. 10 top) – Related to Fig. 1d-1i, the authors state that 15-50% of pYAP expression is preserved in Lats1/Lats2 mutants, how were these values calculated?

Pg. 8. It is interesting that Lats1f/f; Lats2f/f; Yapf/+; Tazf/+; ShhCre/+ mice do not survive after birth considering that they have restored branching and differentiation.

Fig. 1u, 1v should be 1s, 1t, according to the order mentioned in the text.

Pg. 9 – Which figure does “The branching patterns in Lats1f/f; Lats2f/f; SftpcCre/+ (Lats1/2-mosaic) lungs appeared similar to those in control lungs up to 12.5 dpc.” refer to?

Fig. 3g – What are the y-axis units? Should clarify in legend if this is relative length compared to E14.5 controls.

Fig. 3j-q – these figures not mentioned until after Fig. 3p on pg 14

Fig. 3h- axis labels and legend are unreadable

Pg. 10 - What data is “This was confirmed by qPCR analysis” referring to?

Fig. 3i – What are the 3 blue dots representing? Are these targets of YAP?

Supp. Fig. 4 – There is no description of the asterixes and bold text in the legend.

Fig. 4c – it is not mentioned in the legend what blue/red gene names represent.

Fig. 4d, 4h, 6d, 7 – axis labels are impossible to read

Pg. 12 - Fig. 4d not mentioned until pg. 13.

Pg. 16 – This analysis is not shown: “Likewise, we re-analyzed published scRNA-seq of control lungs at 17.5 30 and 18.5 dpc 31 , which showed similar patterns of gene expression to our control lungs.”

Fig. 6 – Perhaps comparing the proportion of cells expressing each marker to the proportion of cells expressing Epcam (or equivalent epithelial marker) would be useful here to compare mutant vs WT.

Fig. 6f/6c – alignment needs fixing

Reviewer #2

(Remarks to the Author)

In this manuscript, Zhang and co-workers examined the function of the Hippo pathway in lung development by inducing loss of LATS1 and LATS2 with different Cre drivers at different times and characterization of the effects of altered LATS activity with cell staining and with extensive single cell sequencing and multiomics.

Previous studies have already shown that the Hippo pathway is important for lung development, and mice mutant for Mst1/2, Lats1/2 and YAP/TAZ have been studied (Mahoney et al., 2014, Dev Cell; Lin et al., 2015 Dev Biol; Lin et al., Elife 2017; Nantie et al 2018 DDevelopment; vanSoldt et al 2019 Development). It is known that Yap is required in the Sox9+ progenitor compartment, but not in the Sox2+ airway compartment, to form distal branches and initiate airway morphogenesis, by using Sox9-Cre and Sox2 Cre drivers (van Soldt et al, 2019 Development). A strength of this manuscript is that they characterize several stages of lung development in LATS mutants using single cell sequencing and use several different Cre drivers. The use of a mosaic Cre is also a potential strength of this paper.

However, the YAP TAZ imaging is poor and needs improvement. In particular they need nuclear segmentation and quantification of YAP/TAZ nuclear staining, not just pYAP staining. Theoretically the mosaic is a powerful approach, however it is underutilized, the mosaics are not shown with informative staining for the Cre, and caveats of 4 floxed alleles are not considered. No information of how they called ATAC-seq peaks in material and methods. Overall better using the Cre reporter data would be more informative. In addition there are several errors in figure citations and legends, and some figures are not clear. Detailed comments listed below.

Fig 1.

-The authors show that loss of LATS 1/2 with SHH cre reduces lung size, mild decrease with mosaic SftpcCre. Mosaic LATS loss with Sftpc Cre led to loss of pYAP, as expected. The authors state that loss of Lats with Sftpc cre leads to reduced Sox9. Why did they not use the tdTomato to define where LATS was lost with the mosaic cre? Is the Sox2 domain expanding? This should be quantified. Also the intermediate domain should be quantified and reference to the recent DevCell paper (March 2025) and discussed.

-They say that lateral sprouting was reduced upon loss of LATS – this is difficult to see and could be quantified.
-They conclude that “ a Sox9 progenitor pool at the most distal subdomain was sufficient to direct overall lung outgrowth though bifurcation”. It is not clear to me that they have shown sufficiency here. It seems necessary re LATS, but if they want to claim Sox9 pool, they should remove Sox9 pool.
-The quantification of wet weight of LATS1/2 Sftpc Cre at e14.5 says no difference to controls, however the figure 1m looks as if the lung is smaller. Perhaps they can replace figure if that representative. Or discuss why there is apparent discrepancy.
-Figure 1 h and i are extremely difficult to see. The authors should do staining to show exactly where the Cre has worked in the mosaic. They also should show YAP/TAZ nuclear staining (not only pYAP).
-Figure 1 says domain branching is affected but bifurcation is normal upon loss of LATS. But on page 44 in their model they say YAP regulates both. Please clarify.

Figure 2.

-Importantly, the authors show that the LATS1/2 ; Shh Cre phenotype on changes of cell types can be suppressed by remove of one copy of YAP and TAZ, supporting a canonical LATS-YAP/TAZ . Some rescue was also seen using Sftpc - Cre.
-The authors should use the modifier “partial” to describe their rescue.
- Cell type counting method for Clara cells- SGCB1A1 is challenging to quantify. It was not clear if they use per nuclei or per length to quantify?
- figure 2a,b,c is not well described and it is not clear what we are seeing in the panels

Figure 3. The authors show that loss of LATS1/2 leads to increased double negative (SOX9-, SOX2-) population . and reduces the SOX2 domain. They suggest that LATS1/2 were deleted from the SOX9 progenitors. They conducted bulk RNAseq and found increased YAP targets, as expected. They also saw reduced expression of distal domain markers and upregulation of pathways involved in the cytoskeleton, ECM, migration as well as reduces markers of conducting airways. EdU incorporation was slightly elevated with expansion of double negative population.

- Figure 3i has typo in that it says it is e14.5 but in the caption and result it says e13.5.
-Would be useful to show in supplemental data the qPCR validation that they said they said they did that confirmed their results.
-Fig 3c and d would be much stronger if the Cre activity marker was shown. The authors should discuss the possibility of incomplete excision of all 4 floxed LATS alleles and how this would affect their interpretation

Figure 4 shows single cell sequencing of lungs at e14.5 in controls and in LATS Sftpc Cre.

-The discussion of the pseudotime was not well described. How does the possibility of several different dosages of LATS affect the interpretation?

Figure 5 shows analysis of LATS loss with Sox9cre and Sftpc Cre and inducible expression to dissect the times. YAP TAZ. Sox9cre and YAP TAZ SftpcCre at e18.5 quantitation

-Need scale bars in c, g, j and k

Figure 6 shows single cell seq of e17.5 control and LATS Sftpc Cre.

The data would be much easier to compare if they showed the control and mutants for each marker (ie, changes in Sox9 in control beside LATS loss). The ATAC seq data is confusingly presented.

Figure 7 shows cluster analysis of mouse and human lung similarity. This could be in the supplemental.

Figure 8 The model is a bit crowded and difficult to read. The authors might want to include here the results of their temporally controlled deletion, and possibly a graphical abstract along the lines

Version 1:

Reviewer comments:

Reviewer #1

(Remarks to the Author)

The authors have performed significant work in addressing our concerns regarding characterization of the mosaic genotype with YAP activity. Approaches to separate mutant and wildtype cells in the scRNA-seq dataset, as well as to validate the separation, are clever and strengthen the analysis. The updated structure of the paper improves the readability. There are two points related to our previous comments and the updated figures should be addressed before publication, to ensure the rigor of the work:

First, the authors should ensure that the processing and labelling of microscopy images are consistent throughout the paper.

Specifically, the labels for pYAP and LATS1 in Supp. Fig. 3a and 3b appear to be reversed, based on signals in the mesenchyme. Additionally, the insets in Supp. Fig. 3i have a different look-up table and xyz position than the main image. Second, the use of 'multilayered epithelia' is not supported. As we noted previously, a multilayered epithelia is a striking phenotype that has not been previously described in the lung and must be substantiated. Comparing Fig. 3v to 3w mosaic and control tissue, we agree that the mosaic epithelia is different in gross morphology than the distal epithelia in the wildtype. However, the mosaic epithelia shows a matched morphology to the more proximal control tissue. Epithelia in the mouse airway is pseudostratified, and nuclear localization alone, as shown in 3v and 3w, is not sufficient to show clear 'multilayering' of epithelia. Additionally, choice of z-section can influence the appearance of epithelia. Images of membrane staining displaying clear multilayered epithelia (apical-basal cell junctions) and neighboring lumen should be shown. We actually recommend removing 'multilayered epithelium' entirely as it does not provide additional value to the discovery. We also have a few minor recommendations to improve the manuscript, but are not critical: First, Figure 3e and 3f are not interpretable and could be removed, because Fig. 3a-3d already makes the relevant points. If kept, contrast should be enhanced.

Second, the summary figures 1p, 2k, 3x, 4n and 5u are redundant as the concepts are presented in Fig. 8a. It is difficult to parse the schematics in Fig. 8b and 8c. We suggest that authors simplify and highlight connection to Hippo signaling and regions of the lung, possibly by expanding 8a and removing 8b,8c.

Reviewer #2

(Remarks to the Author)

Overall this manuscript has improved, but there are still issues that should be addressed before publication.

On the positive side, there have been modifications to the text and figures. Improvements include nicely showing defects in branching (1d), loss of club cell(1f,g) K-M nicely show expansion of the HOPX/ AT1 cells. and loss of AT2 . Fig 2 nicely shows that loss of LATS leads to loss of club cells and this can be partially rescued by loss of one copy of YAP and TAZ

However it still is not clear the mosaic nature of the Sftpc -Cre. Figure 1 does not explain this well. Zoom in of Fig1 a might help explain it. Figure 1 h/l shows sacculae do not form, but arrows would help here to clarify where the sacculae are. Fig 1 NO shows major disruptions of structure, but not clear beyond that. Figure legends should clarify what is occurring.

Fig 3 is still unclear. The pYAP staining in 3e vs 3f does not show clear changes. They state that YAP/TAZ changes do not work (somewhat surprisingly,). If the authors find that this is not doable, an alternative is to do FISH or antibody staining against the YAP target genes that they identified (could be Ajuba, CTGF, or Cyr61). This could also clarify the placement and mutant cells in vivo.

Fig 3S – does show disorganization, but still does not make clear that this is due to loss of domain branching.

Fig 6 and 7 – the scRNAseq data is still not well described, and does not clarify or is overstated. For example, "Our scRNAseq analysis has identified a cell population in which RA signaling controls early lung development". Do the authors mean that they have shown that RA signaling is important re Hippo signaling? I am also unclear if it is true that their study represents " the first report of transitional states from Sox9+ to SOX2+ cells to construct the conducting airway". Or that "Our results also depict the first characterization of transitional states from Sox9+ to AT2 or AT1+ cells to generate alveoli" The current manuscript does not clearly state what was known before their study here and in other points in the text overall..

Version 2:

Reviewer comments:

Reviewer #1

(Remarks to the Author)

I appreciate the changes that the authors have made. They addressed all of my concerns. The simplification of the final figure improves the readability. I would only recommend the authors to do a quick "search and replace" through the manuscript for the word "multilayer". Even though this term has been removed from the main text and replaced by "disorganized epithelium", it was still used in several figure legends.

Reviewer #2

(Remarks to the Author)

The authors have responded well to all my questions, and the manuscript is now substantially improved, and a nice body of work for Nature Communications

Reviewer #1 (Remarks to the Author):

Summary

Zhang et al. investigate the role of Hippo signaling in regulating the expansion and differentiation of the distal epithelial (Sox9+) progenitor domain and neighboring Sox9-/Sox2- in lung development. Although the Hippo pathway has been previously implicated in lung morphogenesis and epithelial fate specification, its precise role in controlling distal epithelial fate transitions remains unclear. In this study, the authors employ mosaic knockout mouse models to selectively disrupt *Lats1/2* and *Yap/Taz*, enabling a heterogeneous perturbation of Hippo signaling. They concluded that hippo signaling regulates epithelial proximal to distal patterning, and further that these perturbations disrupt epithelial cell fate differentiation. The mouse genetic experiments are a tour de force, and the observed differentiation defects are compelling, making this work worthy serious consideration at Nature Communication. However, the manuscript needs some major improvement over the clarity and rigor.

Response:

We thank the reviewer for careful reading, critical insights, and constructive criticisms. We have revised the manuscript in accordance with the reviewer's suggestions. The details of point-by-point responses are below.

Major Concerns

1. Given the mosaic nature of the Cre-mediated recombination, it is unclear whether the epithelial differentiation and lung morphogenesis defects result from the cell-autonomous effect of the knockout or cell non-autonomous effects, making it difficult to interpret the phenotypes. The authors should provide single-cell level characterization of the mosaic lung driven by the SFTPC promoter, to correlate the genotype with phenotype.

a. It is unclear if mutant cells within the mosaic lung have expected Hippo pathway changes. The authors should show the colocalization of Cre recombination (*mTmG* GFP channel) and pYAP to make this point.

Response:

The *ROSA26* locus is purported to contain an open chromatin configuration. As a result, EGFP expression from the *ROSA26^{mTmG}* reporter mice can be induced by low levels of Cre, at which conversion of floxed alleles into null alleles may not occur, as experienced by us and other groups. In the case of *Lats1^{1/f}; Lats2^{2/f}; Sftpc^{Cre/+}; ROSA26^{mTmG/+}* (*Lats1/2*-mosaic) lungs, while all epithelial cells were labeled by EGFP (consistent with broad epithelial Cre expression), removal of epithelial *Lats1/2* was mosaic. Thus, we were unable to use EGFP induction/tdTomato loss as a readout of the conversion of floxed alleles. We have clarified this point in the revised legend of Fig. 1.

As an alternative approach, we used anti-pYAP and anti-LATS1 immunostaining to detect the loss of *Lats1* in *Lats1/2*-mosaic lungs. We found that the LAST1 and pYAP signals were detected only at a subset of the distal lung buds of *Lats1/2*-mosaic lungs (Supplementary Fig. 3). Moreover, the residual LATS1 signal largely coincided with the residual pYAP signal in *Lats1/2*-mosaic lungs (Supplementary Fig. 3). Both the anti-LATS1 and anti-pYAP signals were absent from the more proximal SOX9⁺ subdomain of *Lats1/2*-mosaic lungs (Supplementary Fig. 3). This further supports the notion that the mutant cells (lacking the anti-LATS1 signal) are located at the more proximal SOX9⁺ subdomain.

Together, these studies, together with the mosaic analysis of scRNA-seq data (section (c) below), show that the mutant cells in *Lats1/2*-mosaic lungs exhibit the expected changes in the Hippo pathway.

b. It is unclear how mutant cells were distributed among Sox9+ and Sox2+ domains. This made it difficult to interpret if the outgrowth of the Lats1/2-mosaic lungs was due to compensation by the remaining wildtype cells, cell sorting, or a remaining function of the mutant cells. While these models are difficult to tease apart, a clear characterization of mutant cell position within Sox2/Sox9 domains would help narrowing the likely models. The authors can costain GFP (from the successful recombined mTmG), Sox9, and Sox2. Imaging data should be provided in the figure, rather than just a summary cartoon.

Response:

We have co-stained anti-pYAP/anti-LATS1 (as alluded to above) with anti-SOX9 and anti-SOX2 in *Lats1/2*-mosaic lungs. We found that the residual pYAP/LATS1 signal was detected only at the distal but not the proximal SOX9⁺ subdomain in *Lats1/2*-mosaic lungs (Supplementary Fig. 3). These results suggest that the phenotypes in *Lats1/2*-mosaic lungs were due to the cell-autonomous effects of the mutant cells. This notion is further bolstered by the mosaic analysis of scRNA-seq data described in the next section (c).

c. The analysis on the scRNA-sequencing on mosaic tissue does not separate out mutant and wildtype cells. If feasible, it would be more informative to identify the mutant cells in the data using a combination of Hippo pathway genes (taken together to avoid dropout effects) and provide direct comparison between mutant and wildtype cells.

Response:

As suggested by the reviewer, we have performed additional analysis of scRNA-seq data (Supplementary Fig. 10). We performed label transfer to match cells with transcriptional similarity between control and *Lats1/2*-mosaic (mutant) lungs. This enabled us to identify control and mutant cells in the mutant lungs.

To further confirm the identity of control and mutant cells in the mutant lungs, we examined the expression of YAP/TAZ targets. *Lats1/2* expression in scRNA-seq (GEM-X Universal 3' Gene Expression) cannot be used to distinguish between control and mutant cells; the mutant transcripts were still expressed, and the GEM-X single-cell platform only sequenced the 3' end of the transcript and is unlikely to have sequenced the deleted exons. Thus, we used the expression of canonical YAP/TAZ targets (such as *Ctgf* [*Ccn2*], *Ajuba*, and *Amotl2*) to mark cells that have lost *Lats1/2*. In this case, we found that cells with elevated expression of YAP/TAZ targets correspond to the mutant cells in the mutant lungs identified by label transfer as described above.

We found that mutant cells in the mutant lungs correspond to the new clusters and new transitional states not present in the control lungs. Clusters and transitional states were described in the original submission. This information, together with the immunostaining result of pYAP/LATS1, suggests that the phenotypic consequences are due to the cell-autonomous behavior of mutant cells.

2. Overall, I found the morphological defects in the Lats1/2-mosaic lungs are unsupported or overstated. For example, "multilayer epithelium" is a striking and unusual phenotype, but no imaging data was provided to support this claim. The absence of domain branches in the field of view is difficult to interpret, as it could be due to larger patterning issues that are not specific to the domain branching program. The large-scale airway architecture is constructed by domain branches, which appears generally normal, suggesting the classically defined "domain branching" process is maintained in the mutants. Authors should either avoid using "domain

branching” when discussing the morphological defects, or provide additional morphological analysis to support the defect being specific to a branching mode.

Response:

We apologize for not referring to the images that showed “multilayer epithelium” in the original Fig. 2h and 3k (now Fig. 2h and 1k). This information has been included in the revised legends. We have also added additional images in revised Fig. 3v and 3w to highlight the multilayered epithelium in *Lats1/2*-mosaic lungs.

The morphological defects in *Lats1/2*-mosaic lungs became apparent around 13.5-14.5 *dpc*, by which the large-scale airway architecture is constructed. However, domain branching beyond the initial construction was disrupted in *Lats1/2*-mosaic lungs. This is presented in the revised Fig. 3r and 3t. We acknowledge that a 2D image cannot faithfully capture the 3D distribution of domain branching. To address this issue, we have added quantification of domain branching and bifurcation of the LL1 branch in the revised Fig. 3u.

3. The authors do not untangle the early effects of Lats1/2 on Sox9/Sox2 patterning, and how disruption to this patterning can compound later cell differentiation phenotypes. Specifically, in Fig. 5, the timepoints chosen are too early in development for the motivation authors presented (to separate earlier patterning defects from later cell differentiation phenotypes). The experiment needs to be redone with tamoxifen being dosed just before cell differentiation, which takes place at E17.5. Even with accounting for potential time from dose to recombination, tamoxifen should be applied no earlier than E16. Ideally, authors should also provide a control that shows no major Sox9/Sox2 patterning disruptions just before E17.5 in samples with tamoxifen dosed at their chosen application timepoint. Additionally, it is unclear within the differentiated cells shown to be skewed towards AT1 fate, how many of these are mutant (again because of the mosaic nature of the mouse). Authors could address this by providing the GFP channel (in mTmG mice).

Response:

As suggested by the reviewer, we have performed additional experiments and injected tamoxifen into pregnant female mice that carry control and *Lats1^{ff}; Lats2^{ff}; Sox9^{CreER/+}* embryos at 15.5-16.0 *dpc*. We obtained similar results with an increased ratio of AT1-to-AT2 (revised Fig. 5). Tamoxifen injection at 16.5 *dpc*, when *SOX9^{CreER}* activity is decreasing, did not provide sufficient time for gene inactivation in our experiment, as the lungs were collected at 18.5 *dpc*.

Likewise, we injected tamoxifen into pregnant female mice that carry control and *Yap^{ff}; Taz^{ff}; Sox9^{CreER/+}* embryos at 15.5-16.0 *dpc*. We obtained similar results with an increased ratio of AT2-to-AT1 (revised Fig. 5). Tamoxifen injection at 16.5 *dpc* in this setting was not informative.

As described above, we are unable to use EGFP induction/tdTomato loss from the *ROSA26^{mTmG}* reporter mice as an indicator of conversion of a floxed allele. Our scRNA-seq analysis of control and mutant cells in *Lats1/2*-mosaic lungs, as described above, suggests that the differentiated cells skewed toward the AT1 fate are mutant cells (Supplementary Fig. 10).

4. I found it difficult to match the annotation of the clusters and trajectories in the scRNA-sequencing data analysis with what is previously reported in the field (PMID: 39667932, 33707239). We suggest that authors annotate populations with matched names to previous studies, or provide clear parallels in the text between clusters and previously reported populations.

Response:

As suggested by the reviewer, we have annotated clusters, whenever applicable, with names that match those used in previous studies. PMID: 33707239 annotated populations according to cell type markers in the lung. We have included such annotations in Fig. 6a and Fig. 7a. scRNA-seq data in PMID: 39667932 is derived from what is claimed to be the transitional zone. It has significant expression of *Ager*, *Hopx*, *Lgals3*, and *Sprr1a* in at least two of the three samples, and very few *Sox2*-expressing cells at 14.5 *dpc*, as shown in Supplementary Fig. 8. Thus, it is not appropriate to make a direct comparison between clusters across these datasets.

5. All conclusions made based on lineage and gene regulatory analysis are hypothetical without experimental validation. The authors should be careful with the language used to avoid overstatements.

Response:

We agree with the reviewer that the conclusions from bioinformatics analysis of single-cell data require additional experimental validation, which is beyond the scope of this manuscript. As suggested by the reviewer, we have modified the language in the revised manuscript to avoid overstatements.

6. Human data analysis as presented does not add knowledge to Hippo signaling in Sox9+ progenitors. We suggest that it is removed to streamline the message of the paper.

Response:

In response to the reviewer's comment, we have shortened the description of the human data and moved it to Supplementary Fig. 20 to streamline the message of the paper. However, we have retained the human data, as establishing a connection between the mouse and human scRNA-seq data is important for identifying the conserved mechanisms of lung development.

*7. Currently, the paper structure makes it difficult to identify the key points and often restates ideas out of order. We suggest the following logical sequence to ground the reader in the main message and highlight the impressive results that may otherwise be overlooked: First, characterize the *Lats1/2* mutant with a focus on cell differentiation (the later stages after E17.5), with the striking rescue cell differentiation result as well; Next, work backwards in developmental time and show how *Sox9/Sox2* and transitional progenitor domains are affected; Then ask if cell fate is affected when the *Sox9* domain is not messed up prior (proposed fix to fig 5); Finally, combine and streamline points in Fig 4+6.*

Response:

As suggested by the reviewer, we have reorganized the presentation order of the data. We have presented the overall mutant phenotypes in the conducting airways and sacculles, including the rescue studies, first. This was followed by characterization of changes to the SOX9/SOX2 domains, the experiments on AT1/2 cell fate change, and then single-cell analysis.

Minor Comments

Fig. 1d-1i – It is hard to tell if pYAP levels are actually reduced. Can the authors provide clearer images and potentially quantification

Response:

pYAP levels were significantly reduced or absent in the proximal SOX9⁺ subdomain in the mutant epithelium. We have outlined one branch in Fig. 3e and 3f (the original Fig. 1h and 1i) to facilitate visualization of the changes in pYAP levels. The contrast in pYAP levels was evident

between the epithelium and mesenchyme of the mutant lungs, where pYAP was maintained in the mesenchyme. Additional images of pYAP distribution are available in Supplementary Fig. 3. We have also quantified pYAP (revised Fig. 3i) and found that pYAP was reduced in the proximal but not the distal domain.

Pg. 6 bottom (and pg. 10 top) – Related to Fig. 1d-1i, the authors state that 15-50% of pYAP expression is preserved in Lats1/Lats2 mutants, how were these values calculated?

Response:

Lats1/2 removal by *Sftpc*^{Cre} is mosaic. While loss of *Lats1/2* primarily occurs in the proximal subdomain of SOX9⁺ cells, the extent of *Lats1/2* loss in the distal SOX9⁺ subdomain varies from branch to branch. Hence, the statement that 15-50% of pYAP expression is preserved in *Lats1/2* mutants represents the branch-to-branch variation in *Lats1/2* loss. These values were estimated from the size of the residual SOX9⁺ subdomain, as clarified in the revised legend of Fig. 3.

Pg. 8. It is interesting that Lats1f/f; Lats2f/f; Yapf/+; Tazf/+; ShhCre/+ mice do not survive after birth considering that they have restored branching and differentiation.

Response:

We speculated in the text that perhaps some aspects of pulmonary or extrapulmonary functions were not fully restored.

Fig. 1u, 1v should be 1s, 1t, according to the order mentioned in the text.

Response:

We have reorganized the order of the text description of control and mutant lungs.

Pg. 9 – Which figure does “The branching patterns in Lats1f/f; Lats2f/f; SftpcCre/+ (Lats1/2-mosaic) lungs appeared similar to those in control lungs up to 12.5 dpc.” refer to?

Response:

We have added Supplementary Fig. 5 that shows similar branching patterns between control and *Lats1/2*-mosaic lungs at 12.5 dpc.

Fig. 3g – What are the y-axis units? Should clarify in legend if this is relative length compared to E14.5 controls.

Response:

The y-axis unit represents the relative length, which refers to the relative length between control and mutant lungs, measured in ImageJ. This point has been clarified in the revised figure legend for Fig. 4j (previously Fig. 3g).

Fig. 3j-q – these figures not mentioned until after Fig. 3p on pg 14

Response:

The order of description has been revised.

Fig. 3h- axis labels and legend are unreadable

Response:

We have enlarged the labels and legend in the revised Fig. 4k (previously Fig. 3h).

Pg. 10 - What data is “This was confirmed by qPCR analysis” referring to?

Response:

We have added qPCR data to the revised Fig. 4l.

Fig. 3i – What are the 3 blue dots representing? Are these targets of YAP?

Response:

Fig. 3i is now Fig. 4m. The blue dots represent the most down-regulated genes with expression levels less than 3.5 fold, while the red dots represent the most up-regulated genes with expression levels greater than 3.5 fold. This point has been clarified in the revised figure legend.

Supp. Fig. 4 – There is no description of the asterixes and bold text in the legend.

Response:

We have added a description of asterixis and bold text in the revised legend of Supplementary Fig. 6 (previously Supplementary Fig. 4).

Fig. 4c – it is not mentioned in the legend what blue/red gene names represent.

Response:

Fig. 4c is now Fig. 6c. The blue genes represent those upregulated as cells transition from the left cluster to the right. The red genes are those downregulated as cells transition from the left cluster to the right. This point has been clarified in the revised figure legend.

Fig. 4d, 4h, 6d, 7 – axis labels are impossible to read

Response:

We have enlarged the labels on the axes in revised Fig. 6d, 6h, 7d, 7h (previously Fig. 4d, 4h, 6d, 6h).

Pg. 12 - Fig. 4d not mentioned until pg. 13.

Response:

scRNA-seq data for the control lungs were grouped for consistency. Thus, Fig. 4d (now Fig. 6d) was mentioned later in the text.

Pg. 16 – This analysis is not shown: “Likewise, we re-analyzed published scRNA-seq of control lungs at 17.5 30 and 18.5 dpc 31 , which showed similar patterns of gene expression to our control lungs.”

Response:

We have included the re-analysis of published scRNA-seq data from mouse lungs at different developmental stages as supplementary figures (Supplementary Figs. 7, 8, 14, and 15).

Fig. 6 – Perhaps comparing the proportion of cells expressing each marker to the proportion of cells expressing Epcam (or equivalent epithelial marker) would be useful here to compare mutant vs WT.

Response:

We have included this information in the revised Supplementary Fig. 9.

Fig. 6f/6c – alignment needs fixing

Response:

We have fixed the alignment in the revised Fig. 7c and 7g (previously Fig. 6c and 6f).

Reviewer #2 (Remarks to the Author):

In this manuscript, Zhang and co-workers examined the function of the Hippo pathway in lung development by inducing loss of LATS1 and LATS2 with different Cre drivers at different times and characterization of the effects of altered LATS activity with cell staining and with extensive single cell sequencing and multiomics.

Previous studies have already shown that the Hippo pathway is important for lung development, and mice mutant for Mst1/2, Lats1/2 and YAP/TAZ have been studied (Mahoney et al., 2014, Dev Cell; Lin et al., 2015 Dev Biol; Lin et al., Elife 2017; Nantie et al 2018 DDevelopment; vanSoldt et al 2019 Development). It is known that Yap is required in the Sox9+ progenitor compartment, but not in the Sox2+ airway compartment, to form distal branches and initiate airway morphogenesis, by using Sox9-Cre and Sox2 Cre drivers (van Soldt et al, 2019 Development). A strength of this manuscript is that they characterize several stages of lung development in LATS mutants using single cell sequencing and use several different Cre drivers. The use of a mosaic Cre is also a potential strength of this paper.

However, the YAP TAZ imaging is poor and needs improvement. In particular they need nuclear segmentation and quantification of YAP/TAZ nuclear staining, not just pYAP staining. Theoretically the mosaic is a powerful approach, however it is underutilized, the mosaics are not shown with informative staining for the Cre, and caveats of 4 floxed alleles are not considered. No information of how they called ATAC-seq peaks in material and methods. Overall better using the Cre reporter data would be more informative. In addition there are several errors in figure citations and legends, and some figures are not clear. Detailed comments listed below.

Response:

We thank the reviewer for many important insights, suggestions, and comments. We have revised the manuscript accordingly and addressed the concerns raised by the reviewer. The details of point-by-point responses are below.

Fig 1.

-The authors show that loss of LATS ½ with SHH cre reduces lung size, mild decrease with mosaic SftpcCre. Mosaic LATS loss with Sftpc Cre led to loss of pYAP, as expected. The authors state that loss of Lats with Sftpc cre leads to reduced Sox9. Why did they not use the tdTomato to define where LATS was lost with the mosaic cre? Is the Sox2 domain expanding? This should be quantified. Also the intermediate domain should be quantitated and reference to the recent DevCell paper (March 2025) and discussed.

Response:

The ROSA26 locus is purported to contain an open chromatin configuration. As a result, EGFP expression from the ROSA26^{mTmG} reporter mice can be induced by low levels of Cre, at which conversion of floxed alleles into null alleles may not occur, as experienced by us and other groups. In the case of *Lats1^{fl/fl}; Lats2^{fl/fl}; Sftpc^{Cre/+}; ROSA26^{mTmG/+}* lungs, while all epithelial cells were labeled by EGFP (consistent with broad epithelial Cre expression), removal of *Lats1/2* was mosaic. Thus, we were unable to use EGFP induction/TdTomato loss as a readout of the conversion of floxed alleles. We have clarified this point in the revised legend of Fig. 1.

Instead, we have used anti-LATS1 and anti-pYAP to indicate where *Lats1/2* were lost. We found that the anti-LATS1 signal largely coincided with the anti-pYAP signal at a subset of the distal SOX9⁺ subdomain (Supplementary Fig. 3). Both the anti-LATS1 and anti-pYAP signals were absent from the more proximal SOX9⁺ subdomain. This further supports the notion that the

mutant cells (lacking the anti-LATS1 signal) are located at the more proximal SOX9⁺ subdomain.

Removal of *Lats1/2* in the distal SOX9⁺ cells in *Lats1/2*-mosaic lungs is mosaic. This enables the production of some SOX2⁺ cells, which abut the distal SOX9⁺ cells. As a result, when one examines the distribution of SOX2⁺ cells along the proximo-distal axis, the SOX2⁺ domain appears to be expanding in *Lats1/2*-mosaic lungs from 14.5 to 16.5 *dpc*. However, the majority of the proximal region is devoid of SOX2⁺ cells except for the region immediately adjacent to the SOX9⁺ domain. The absolute size of the SOX2⁺ domain is reduced in *Lats1/2*-mosaic lungs compared to control lungs. We have quantified the SOX2⁺ domain in the revised Fig. 4i.

Quantification of the intermediate domain is shown in Fig. 3g of the original application (the revised Fig. 4j).

We have referenced the recent *Dev Cell* paper (March 2025) in the revised manuscript. We have also provided a re-analysis of published scRNA-seq data at different developmental stages, including that in the recent *Dev Cell* paper, in Supplementary Figs. 7, 8, 14, and 15. scRNA-seq data in the recent *Dev Cell* paper (March 2025) is derived from what is claimed to be the transitional zone. It has significant expression of *Ager*, *Hopx*, *Lgals3*, and *Sprr1a* in at least two of the three samples, and very few *Sox2*-expressing cells at 14.5 *dpc*, as shown in Supplementary Fig. 8. Thus, it is not appropriate to make a direct comparison between clusters across these datasets.

-They say that lateral sprouting was reduced upon loss of LATS – this is difficult to see and could be quantified.

Response:

As suggested by the reviewer, we have quantified the lateral sprouting (domain branching) in the revised Fig. 3u.

-They conclude that “ a Sox9 progenitor pool at the most distal subdomain was sufficient to direct overall lung outgrowth through bifurcation”. It is not clear to me that they have shown sufficiency here. It seems necessary re LATS, but if they want to claim Sox9 pool, they should remove Sox9 pool.

Response:

When the non-distal SOX9⁺ pool was removed by *Sftpc*^{Cre}, lung outgrowth was largely preserved. By contrast, when the entire SOX9⁺ pool was removed by *Shh*^{Cre}, lung outgrowth was blocked. Together, these observations suggest that the most distal subdomain was sufficient to direct overall lung outgrowth through bifurcation. We have clarified this point in the revised text. This point is also illustrated in the schematic diagram in Fig. 8b.

-The quantification of wet weight of LATS1/2 Sftpc Cre at e14.5 says no difference to controls, however the figure 1m looks as if the lung is smaller. Perhaps they can replace figure if that representative. Or discuss why there is apparent discrepancy.

Response:

Fig. 1m (now Fig. 3k) is representative. *Lats1/2*-mosaic (mutant) lungs were slightly smaller but denser. No difference in proliferation was noted at 14.5 *dpc* (Supplementary Fig. 4). Together, the wet weights were similar between the control and mutant lungs. This point has been clarified in the revised legend of Fig. 1.

-Figure 1 h and i are extremely difficult to see. The authors should do staining to show exactly where the Cre has worked in the mosaic. They also should show YAP/TAZ nuclear staining (not only pYAP).

Response:

pYAP levels were significantly reduced or absent in the proximal SOX9⁺ subdomain in the lung epithelium of *Lats1/2*-mosaic mice. We have outlined one branch in the revised Fig. 3e and 3f (the original Fig. 1h and 1i) to facilitate visualization of the changes in pYAP levels. The contrast in pYAP levels was evident between the epithelium and mesenchyme of the mutant lungs, where pYAP was maintained in the mesenchyme. Additional images of pYAP distribution are available in Supplementary Fig. 3.

We have performed YAP staining and nuclear YAP was present in epithelial cells of lung buds lacking the LATS1 signal (Supplementary Fig. 3). However, nuclear YAP was also detected in cells in the distal SOX9⁺ subdomain that retained the LATS1 signal (Supplementary Fig. 3). Immunostaining of YAP did not reveal an apparent difference in nuclear YAP intensity between cells with and without LATS1. Thus, *Lats1/2*-deficient cells in *Lats1/2*-mosaic lungs could be identified by reduced pYAP via immunostaining (Fig. 3, Supplementary Fig. 3) and activation of YAP targets by scRNA-seq (Supplementary Fig. 10); however, their changes in nuclear YAP were not apparent at 14.5 *dpc*. This may reflect the dynamic shuttling of YAP between the cytoplasm and the nucleus in lung epithelial cells. Moreover, the presence of nuclear YAP in control cells is known to depend on the developmental stage and cell type.

-Figure 1 says domain branching is affected but bifurcation is normal upon loss of LATS. But on page 44 in their model they say YAP regulates both. Please clarify.

Response:

The removal of *Lats1/2* by *Sftpc*^{Cre} is mosaic. Since the distal SOX9⁺ subdomain was largely spared, bifurcation was unaffected. In contrast, removal of *Lats1/2* by *Shh*^{Cre} affected both domain branching and bifurcation. As a result, in our model, YAP was depicted to affect both domain branching and bifurcation. This point has been clarified in the revised legend of Fig. 8.

Figure 2.

-Importantly, the authors show that the LATS1/2 ; Shh Cre phenotype on changes of cell types can be suppressed by remove of one copy of YAP and TAZ, supporting a canonical LATS-YAP/TAZ . Some rescue was also seen using Sftpc -Cre.

-The authors should use the modifier "partial" to describe their rescue.

Response:

As suggested by the reviewer, we have added the modifier "partial" to describe the rescue in the revised text.

- Cell type counting method for Clara cells- SGCB1A1 is challenging to quantify. It was not clear if they use per nuclei or per length to quantify?

Response:

We used nuclei for cell counting. This point has been clarified in the revised legend of Fig. 2.

- figure 2a,b,c is not well described and it is not clear what we are seeing in the panels

Response:

We have added descriptions to the revised legend for Fig. 2a, 2b, and 2c to clarify what is presented in these panels (e.g., surface views of the distal lungs where only a few large, thickened stalks were observed in *Lats1/2*-mosaic lungs in 2b).

Figure 3. The authors show that loss of LATS1/2 leads to increased double negative (SOX9-, SOX2-) population . and reduces the SOX2 domain. They suggest that LATS1/2 were deleted from the SOX9 progenitors. They conducted bulk RNAseq and found increased YAP targets, as expected. They also saw reduced expression of distal domain markers and upregulation of pathways involved in the cytoskeleton, ECM, migration as well as reduces markers of conducting airways. EdU incorporation was slightly elevated with expansion of double negative population.

- *Figure 3i has typo in that it says it is e14.5 but in the caption and result it says e13.5.*

Response:

We have corrected the inconsistency in the revised Fig. 4 (previously Fig. 3). Both are now 13.5 dpc.

- *Would be useful to show in supplemental data the qPCR validation that they said they said they did that confirmed their results.*

Response:

We have added qPCR data to the revised Fig. 4i.

- *Fig 3c and d would be much stronger if the Cre activity marker was shown. The authors should discuss the possibility of incomplete excision of all 4 floxed LATS alleles and how this would affect their interpretation*

Response:

From genotyping and phenotypic analysis, we found that the loss of one, two, or three alleles of *Lats1/2* by *Sftpc^{Cre}* or *Shh^{Cre}* did not cause lung defects. We have clarified this point in the revised legend of Fig. 1.

Figure 4 shows single cell sequencing of lungs at e14.5 in controls and ing LATS Sftpc Cre.

- *The discussion of the pseudotime was not well described. How does the possibility of several different dosages of LATS affect the interpretation?*

Response:

We have provided a better description of pseudotime.

As described above, loss of one, two, or three alleles of *Lats1/2* by *Sftpc^{Cre}* or *Shh^{Cre}* did not cause lung defects. Thus, the mutant lungs used for single-cell analysis contained mutant cells that had lost all four alleles of *Lats1/2*. We have clarified this point in the revision.

Figure 5 shows analysis of LATS loss with Sox9cre and Sftpc Cre and inducible expression to dissect the times. YAP TAZ. Sox9cre and YAP TAZ SftpcCre at e18.5 quantitation

- *Need scale bars in c, g, j and k*

Response:

As indicated in the original figure legend, the scale bar in Fig. 5b was for Fig. 5b, 5c, 5j, 5k, and the scale bar in Fig. 5f was for Fig. 5f, 5g. To avoid confusion, we have added scale bars to multiple panels of the revised Fig. 6 (previously Fig. 5).

Figure 6 shows single cell seq of e17.5 control and LATS Sftpc Cre. The data would be much easier to compare if they showed the control and mutants for each marker (ie, changes in Sox9 in control beside LATS loss). The ATAC seq data is confusingly presented.

Response:

Marker expression panels for either control or mutant lungs were grouped for consistency and to avoid confusion by the readers.

We have modified the presentation of snATAC-seq in the revised Supplementary Figs. 13 and 19.

Figure 7 shows cluster analysis of mouse and human lung similarity. This could be in the supplemental.

Response:

We have moved the cluster analysis of mouse and human lungs in the original Fig. 7 to Supplementary Fig. 20.

Figure 8 The model is a bit crowded and difficult to read. The authors might want to include here the results of their temporally controlled deletion, and possibly a graphical abstract along the lines

Response:

As suggested by the reviewer, we have modified the model to include the results of the temporal deletion of Hippo pathway components. We have also added a graphical abstract (Fig. 8a) to Fig. 8.

Reviewer #1 (Remarks to the Author):

The authors have performed significant work in addressing our concerns regarding characterization of the mosaic genotype with YAP activity. Approaches to separate mutant and wildtype cells in the scRNA-seq dataset, as well as to validate the separation, are clever and strengthen the analysis. The updated structure of the paper improves the readability. There are two points related to our previous comments and the updated figures should be addressed before publication, to ensure the rigor of the work:

First, the authors should ensure that the processing and labelling of microscopy images are consistent throughout the paper. Specifically, the labels for pYAP and LATS1 in Supp. Fig. 3a and 3b appear to be reversed, based on signals in the mesenchyme. Additionally, the insets in Supp. Fig. 3i have a different look-up table and xyz position than the main image.

Response:

We thank the reviewer for pointing out our mistake in labeling pYAP and LATS1 in the original Supplementary Fig. 3a and 3b (now 3g and 3h). While ensuring consistent pseudo-colors for a given antibody, we accidentally swapped the labels for pYAP and LATS1. This has been corrected in the revised Supplementary Figs. 3g and 3h.

Regarding the insets in Supplementary Fig. 3, they are confocal images from adjacent sections; therefore, their xyz positions differ slightly from those of the main image taken with a fluorescence microscope. These images have been replaced by new data in the revised Supplementary Fig. 3 as described below.

Our original YAP immunostaining utilized a rabbit anti-YAP antibody. Since the anti-LATS1 antibody was also produced in rabbit, we had to rely on the FlexAble Antibody Labeling kit for co-immunostaining. We suspect that this has reduced both the YAP and LATS1 signals. To resolve this issue, we used a mouse anti-YAP antibody for co-immunostaining with anti-LATS1. We found that YAP signals were stronger in the proximal region where LATS1 was removed than in the distal region where LATS1 was preserved. We have replaced the old data with the new ones (revised Supplementary Fig. 3o-r).

Second, the use of ‘multilayered epithelia’ is not supported. As we noted previously, a multilayered epithelia is a striking phenotype that has not been previously described in the lung and must be substantiated. Comparing Fig. 3v to 3w mosaic and control tissue, we agree that the mosaic epithelia is different in gross morphology than the distal epithelia in the wildtype. However, the mosaic epithelia shows a matched morphology to the more proximal control tissue. Epithelia in the mouse airway is pseudostratified, and nuclear localization alone, as shown in 3v and 3w, is not sufficient to show clear ‘multilayering’ of epithelia. Additionally, choice of z-section can influence the appearance of epithelia. Images of membrane staining displaying clear multilayered epithelia (apical-basal cell junctions) and neighboring lumen should be shown. We actually recommend removing ‘multilayered epithelium’ entirely as it does not provide additional value to the discovery.

Response:

As suggested by the reviewer, we have removed the wording of “multilayered epithelium” in the revision. We also added speculation on how the disorganized epithelium in *Lats1/2*-mosaic lungs may have arisen in the revised legend of Fig. 1.

We also have a few minor recommendations to improve the manuscript, but are not critical: First, Figure 3e and 3f are not interpretable and could be removed, because Fig. 3a-3d already makes the relevant points. If kept, contrast should be enhanced.

Response:

As suggested by the reviewer, we have enhanced the contrast of Figs. 3e and 3f to make the residual pYAP more visible in the mutant lung.

We have added images of whole-mount immunostaining of pYAP and Ecad at 13.5 *dpc* to the revised Supplementary Fig. 3 (a-f) to further substantiate the point of residual pYAP in *Lats1/2*-mosaic lungs.

Second, the summary figures 1p, 2k, 3x, 4n and 5u are redundant as the concepts are presented in Fig. 8a. It is difficult to parse the schematics in Fig. 8b and 8c. We suggest that authors simplify and highlight connection to Hippo signaling and regions of the lung, possibly by expanding 8a and removing 8b,8c.

Response:

As suggested by the reviewer, we have removed the summary figures 1p and 3x. We have also merged summary figures 4n and 5u into the revised Fig. 8.

We have removed Figs. 8b and 8c in the original Fig. 8 and placed them in a supplementary figure (Supplementary Fig. 22) for the interested readers. We have expanded the original Fig. 8a to Fig. 8a-d.

Reviewer #2 (Remarks to the Author):

Overall this manuscript has improved, but there are still issues that should be addressed before publication.

On the positive side, there have been modifications to the text and figures. Improvements include nicely showing defects in branching (1d), loss of club cell (1f,g) K-M nicely show expansion of the HOPX/ AT1 cells. and loss of AT2 . Fig 2 nicely shows that loss of LATS leads to loss of club cells and this can be partially rescued by loss of one copy of YAP and TAZ

*However it still is not clear the mosaic nature of the *Sftpc*-Cre. Figure 1 does not explain this well. Zoom in of Fig1 a might help explain it. Figure 1 h/l shows sacculae do not form, but arrows would help here to clarify where the sacculae are. Fig 1 NO shows major disruptions of structure, but not clear beyond that. Figure legends should clarify what is occurring.*

Response:

The mosaic nature of *Sftpc*^{Cre} originates from its selective (regional) effects on converting floxed alleles of *Lats1/2* into null alleles. The mosaic nature of *Sftpc*^{Cre} does not come from selective (regional) expression of *Sftpc*^{Cre}. Consistent with this notion, induction of GFP by *Sftpc*^{Cre} is ubiquitous in Fig. 1a. We have added more notes to clarify this point in the revised legend of Fig. 1.

As suggested by the reviewer, we have labeled the open and closed sacculae in the revised Fig. 1h and added a description of them in the revised legend of Fig. 1.

As suggested by the reviewer, we have also expanded the descriptions for 1n and 1o in the revised legend of Fig. 1

Fig 3 is still unclear. The pYAP staining in 3e vs 3f does not show clear changes. They state that YAP/TAZ changes do not work (somewhat surprisingly,). If the authors find that this is not doable, an alternative is to do FISH or antibody staining against the YAP target genes that they identified (could be Ajuba, CTGF, or Cyr61). This could also clarify the placement and mutant cells in vivo.

Response:

We have enhanced the contrast of Fig. 3e and 3f to make the residual pYAP more visible in the mutant lung.

We have added images of whole-mount immunostaining of pYAP and Ecad at 13.5 dpc to the revised Supplementary Fig. 3 (a-f) to further substantiate the point of residual pYAP in *Lats1/2*-mosaic lungs.

Our original YAP immunostaining utilized a rabbit anti-YAP antibody. Since the anti-LATS1 antibody was also produced in rabbit, we had to rely on the FlexAble Antibody Labeling kit for co-immunostaining. We suspect that this has reduced both the YAP and LATS1 signals. To resolve this issue, we used a mouse anti-YAP antibody for co-immunostaining with anti-LATS1. We found that YAP signals were stronger in the proximal region where LATS1 was removed than in the distal region where LATS1 was preserved. We have replaced the old data with the new ones (revised Supplementary Fig. 3o-r).

As suggested by the reviewer, we have also performed immunostaining using antibodies against CTGF. We detected CTGF expression in regions where LATS1 was absent (revised Supplementary Fig. 3s, 3t)

Fig 3S – does show disorganization, but still does not make clear that this is due to loss of domain branching.

Response:

The boxed area in Fig. 3s contains daughter branches of the L.L1 branch. Domain branching of L.L1 (Metzger et al. 2008) gives rise to lung branches along the stalk and epithelial buds near the surface in control lungs (see Figs. 3q, 3r). Fig. 3s showed loss or reduced epithelial buds near the surface that are derived from the L.L1 branch in *Lats1/2*-mosaic lungs. Fig. 3s also exhibited loss or reduced branches along the stalk of the LL1 branch. These results are consistent with defective domain branching of the L.L1 branch in *Lats1/2*-mosaic lungs. These points have been clarified in the revised legend of Fig. 3

Fig 6 and 7 – the scRNAseq data is still not well described, and does not clarify or is overstated. For example, "Our scRNAseq analysis has identified a cell population in which RA signaling controls early lung development". Do the authors mean that they have shown that RA signaling is important re Hippo signaling?

Response:

As suggested by the reviewer, we have added additional qualifiers to the genes and pathways identified by scRNA-seq. For example, we revised our statement to "Our scRNA-seq analysis has identified a candidate cell population in which RA signaling controls early development".

I am also unclear if it is true that their study represents " the first report of transitional states from Sox9+ to SOX2+ cells to construct the conducting airway". Or that "Our results also depict the first characterization of transitional states from Sox9+ to AT2 or AT1+ cells to generate alveoli" The current manuscript does not clearly state what was known before their study here and in other points in the text overall..

Response:

We have added a qualifier and used "first extensive" instead of "first" in the revision.

As noted in our prior response, it is not appropriate to compare scRNA-seq data from the *Dev Cell* paper (March 2025) with other datasets, including ours. scRNA-seq data in the *Dev Cell* paper (March 2025) was derived from the "transition zone" and contains few SOX2⁺ cells. It does not allow the tracing of the SOX9–SOX2 transition. Moreover, as noted in the legend of Supplementary Fig. 8, the scRNA-seq data in the *Dev Cell* paper (March 2025) include three samples, two of which exhibit markers of later stages of lung development, which makes interpretation challenging. Thus, at face value, our studies represent the "first extensive", if not the "first", report or characterization of the transition of SOX9⁺ cells to SOX2⁺ cells and to AT1/2 cells.